# One-Step Effective Diffusion Network for Real-World Image Super-Resolution

**Rongyuan Wu**[1,2,*]**, Lingchen Sun**[1,2,*]**, Zhiyuan Ma**[1,*]**, Lei Zhang**[1,2,†]

[1]The Hong Kong Polytechnic University      [2]OPPO Research Institute

{rong-yuan.wu, ling-chen.sun, zm2354.ma}@connect.polyu.hk, cslzhang@comp.polyu.edu.hk

[*]Equal contribution      [†]Corresponding author

## Abstract

The pre-trained text-to-image diffusion models have been increasingly employed to tackle the real-world image super-resolution (Real-ISR) problem due to their powerful generative image priors. Most of the existing methods start from random noise to reconstruct the high-quality (HQ) image under the guidance of the given low-quality (LQ) image. While promising results have been achieved, such Real-ISR methods require multiple diffusion steps to reproduce the HQ image, increasing the computational cost. Meanwhile, the random noise introduces uncertainty in the output, which is unfriendly to image restoration tasks. To address these issues, we propose a one-step effective diffusion network, namely OSEDiff, for the Real-ISR problem. We argue that the LQ image contains rich information to restore its HQ counterpart, and hence the given LQ image can be directly taken as the starting point for diffusion, eliminating the uncertainty introduced by random noise sampling. We finetune the pre-trained diffusion network with trainable layers to adapt it to complex image degradations. To ensure that the one-step diffusion model could yield HQ Real-ISR output, we apply variational score distillation in the latent space to conduct KL-divergence regularization. As a result, our OSEDiff model can efficiently and effectively generate HQ images in just one diffusion step. Our experiments demonstrate that OSEDiff achieves comparable or even better Real-ISR results, in terms of both objective metrics and subjective evaluations, than previous diffusion model-based Real-ISR methods that require dozens or hundreds of steps. The source codes are released at https://github.com/cswry/OSEDiff.

## 1 Introduction

Image super-resolution (ISR) [13, 66, 29, 65, 6, 24, 46, 27, 61] is a classical yet still active research problem, which aims to restore a high-quality (HQ) image from its low-quality (LQ) observation suffering from degradations of noise, blur and low-resolution, *etc*. While one line of ISR research [13, 66, 29, 65, 6] simplifies the degradation process from HQ to LQ images as bicubic downsampling (or downsampling after Gaussian blur) and focus on the study on network architecture design, the trained models can hardly be generalized to real-world LQ images, whose degradations are often unknown and much more complex. Therefore, another increasingly popular line of ISR research is the so-called real-world ISR (Real-ISR) [61, 45] problem, which aims to reproduce perceptually realistic HQ images from the LQ images captured in real-world applications.

There are two major issues in training a Real-ISR model. One is how to build the LQ-HQ training image pairs, and another is how to ensure the naturalness of restored images, *i.e.*, how to ensure that the restored images follow the distribution of HQ natural images. For the first issue, some researchers

---

This work is supported by the PolyU-OPPO Joint Innovation Lab.

38th Conference on Neural Information Processing Systems (NeurIPS 2024).

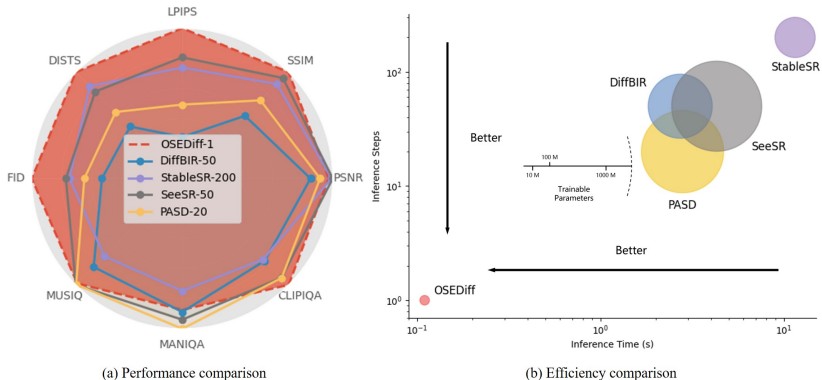

(a) Performance comparison          (b) Efficiency comparison

Figure 1: Performance and efficiency comparison among SD-based Real-ISR methods. (a). Performance comparison on the DrealSR benchmark [51]. Metrics like LPIPS and NIQE, where smaller scores indicate better image quality, are inverted and normalized for display. OSEDiff achieves leading scores on most metrics with only one diffusion step. (b). Model efficiency comparison. The inference time is tested on an A100 GPU with $512 \times 512$ input image size. OSEDiff has the fewest trainable parameters and is over 100 times faster than StableSR [42].

have proposed to collect real-world LQ-HQ image pairs using long-short camera focal lenses [3, 51]. However, this is very costly and can only cover certain types of real-world image degradations. Another more economical way is to simulate the real-world LQ-HQ image pairs by using complex image degradation pipelines. The representative works include BSRGAN [61] and Real-ESRGAN [45], where a random shuffling of basic degradation operators and a high-order degradation model are respectively used to generate LQ-HQ image pairs.

With the given training data, how to train a robust Real-ISR model to output perceptually natural images with high quality becomes a key issue. Simply learning a mapping network between LQ-HQ paired data with pixel-wise losses can lead to over-smoothed results [24, 46]. It is crucial to integrate natural image priors into the learning process to reproduce HQ images. A few methods have been proposed to this end. The perceptual loss [18] explores the texture, color, and structural priors in a pre-trained model such as VGG-16 [38] and AlexNet [23]. The generative adversarial networks (GANs) [14] alternatively train a generator and a discriminator, and they have been adopted for Real-ISR tasks [24, 46, 45, 61, 27, 53]. The generator network aims to synthesize HQ images, while the discriminator network aims to distinguish whether the synthesized image is realistic or not. While great successes have been achieved, especially in the restoration of specific classes of images such as face images [56, 44], GAN-based Real-ISR tends to generate unpleasant details due to the unstable adversarial training and the difficulties in discriminating the image space of diverse natural scenes.

The recently developed generative diffusion models (DM) [39, 16], especially the large-scale pre-trained text-to-image (T2I) models [37, 36], have demonstrated remarkable performance in various downstream tasks. Having been trained on billions of image-text pairs, the pre-trained T2I models possess powerful natural image priors, which can be well exploited to improve the naturalness and perceptual quality of Real-ISR outputs. Some methods [42, 57, 31, 52, 40, 59] have been developed to employ the pre-trained T2I model for solving the Real-ISR problem. While having shown impressive results in generating richer and more realistic image details than GAN-based methods, the existing SD-based methods have several problems to be further addressed. First, these methods typically take random Gaussian noise as the start point of the diffusion process. Though the LQ images are used as the control signal with a ControlNet module [63], these methods introduce unwanted randomness in the output HQ images [40]. Second, the restored HQ images are usually obtained by tens or even hundreds of diffusion steps, making the Real-ISR process computationally expensive. Though some one-step diffusion based Real-ISR methods [48] have been recently proposed, they fail in achieving high-quality details compared to multi-step methods.

To address the aforementioned issues, we propose a **O**ne-**S**tep **E**ffective **Diff**usion network, **OSEDiff** in short, for the Real-ISR problem. The UNet backbone in pre-trained SD models has strong capability to transfer the input data into another domain, while the given LQ image actually has rich information to restore its HQ counterpart. Therefore, we propose to directly feed the LQ images into the pre-trained SD model without introducing any random noise. Meanwhile, we integrate trainable

LoRA layers [17] into the pre-trained UNet, and finetune the SD model to adapt it to the Real-ISR task. On the other hand, to ensure that the one-step model can still produce HQ natural images as the multi-step models, we utilize variational score distillation (VSD) [49, 58, 10] for KL-divergence regularization. This operation effectively leverages the powerful natural image priors of pre-trained SD models and aligns the distribution of generated images with natural image priors. As illustrated in Fig. 1, our extensive experiments demonstrate that OSEDiff achieves comparable or superior performance measures to state-of-the-art SD-based Real-ISR models, while it significantly reduces the number of inference steps from $N$ to 1 and has the fewest trainable parameters, leading to more than $\times 100$ speedup in inference time over previous methods such as StableSR [42].

## 2   Related Work

Starting from SRCNN [13], deep learning-based methods have become prevalent for ISR. A variety of methods have been proposed [30, 66, 67, 9, 5, 29, 65, 6, 7] to improve the accuracy of ISR reconstruction. However, most of these methods assume simple and known degradations such as bicubic downsampling, limiting their applications to real-world images with complex and unknown degradations. In recent years, researches have been exploring the potentials of generative models, including GAN [14] and diffusion networks [16], for solving the Real-ISR problem.

**GAN-based Real-ISR.** The use of GAN for photo-realistic ISR can be traced back to SRGAN [24], where the image degradation is assumed to be bicubic downsampling. Later on, researchers found that GAN has the potential to perform real-world image restoration with more complex degradations [61, 45]. Specifically, by using randomly shuffled degradation and high-order degradation to generate more realistic LQ-HQ training pairs, BSRGAN [61] and Real-ESRGAN [45] demonstrate promising Real-ISR results, which trigger many following works [4, 27, 28, 53]. DASR [28] designs a tiny network to predict the degradation parameters to handle degradations of various levels. SwinIR [29] switches the generator from CNNs to stronger transformers, further enhancing the performance of Real-ISR. However, the adversarial training process in GAN is unstable and its discriminator is limited in telling the quality of diverse natural image contents. Therefore, GAN-based Real-ISR methods often suffer from unnatural visual artifacts. Some works such as LDL [27] and DeSRA [53] can suppress much the artifacts, yet they are difficult to generate more natural details.

**Diffusion-based Real-ISR.** Some early attempts [21, 20, 47] employ the denoising diffusion probabilistic models (DDPMs) [16, 39, 11] to address the ISR problem by assuming simple and known degradations (*e.g.*, bicubic downsampling). These methods are training-free by modifying the reverse transition of pre-trained DDPMs using gradient descent, but they cannot be applied to complex unknown degradations. Recent researches [42, 57, 31, 40, 59] have leveraged stronger pre-trained T2I models, such as Stable Diffusion (SD) [1], to tackle the Real-ISR problem. In general, they introduce an adapter [63] to fine-tune the SD model to reconstruct the HQ image with the LQ image as the control signal. StableSR [42] finetunes a time-aware encoder and employs feature warping to balance fidelity and perceptual quality. PASD [57] extracts both low-level and high-level features from the LQ image and inputs them to the pre-trained SD model with a pixel-aware cross attention module. To further enhance the semantic-aware ability of the Real-ISR model, SeeSR [52] introduces degradation-robust tag-style text prompts and utilizes soft prompts to guide the diffusion process. To mitigate the potential risks of diffusion uncertainty, CCSR [40] leverages a truncated diffusion process to recover structures and finetunes the VAE decoder by adversarial training to enhance details. SUPIR [59] leverages the powerful generation capability of SDXL model and the strong captioning capability of LLaVA [32] to synthesize rich image details.

The above mentioned methods, however, require tens or even hundreds of steps to complete the diffusion process, resulting in unfriendly latency. SinSR shortens ResShift [60] to a single-step inference by consistency preserving distillation. Nevertheless, the non-distrbution-based distillation loss tends to obtain smooth results, and the model capacity of SinSR and ResShift are much smaller than the SD models to address Real-ISR problems.

## 3   Methodology

### 3.1   Problem Modeling

Real-ISR is to estimate an HQ image $\hat{x}_H$ from the given LQ image $x_L$. This task can be conventionally modeled as an optimization problem: $\hat{x}_H = \mathrm{argmin}_{x_H}(\mathcal{L}_{data}(\Phi(x_H), x_L) + \lambda \mathcal{L}_{reg}(x_H))$, where $\Phi$ is the degradation function, $\mathcal{L}_{data}$ is the data term to measure the fidelity of the optimization

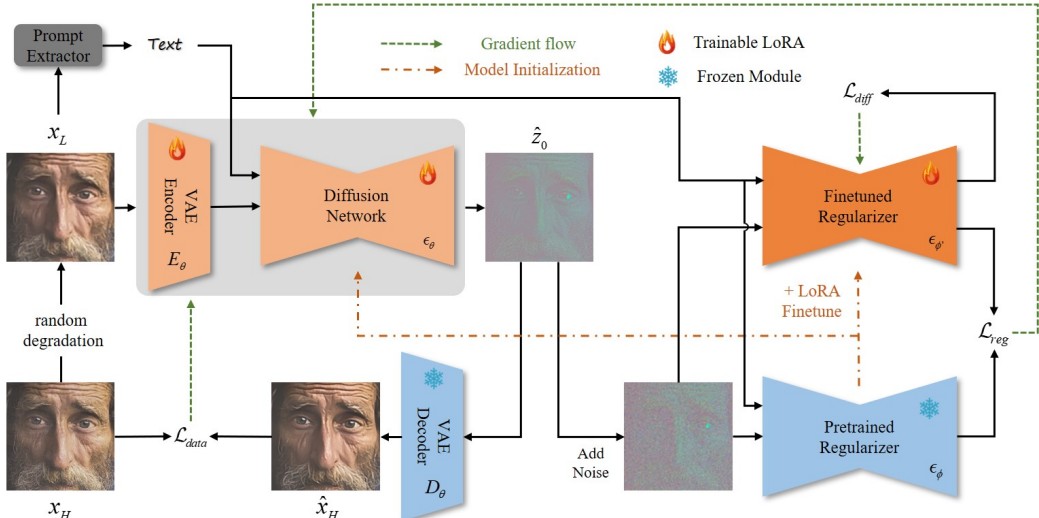

Figure 2: The training framework of OSEDiff. The LQ image is passed through a trainable encoder $E_\theta$, a LoRA finetuned diffusion network $\epsilon_\theta$ and a frozen decoder $D_\theta$ to obtain the desired HQ image. In addition, text prompts are extracted from the LQ image and input to the diffusion network to stimulate its generation capacity. Meanwhile, the output of the diffusion network $\epsilon_\theta$ will be sent to two regularizer networks (a frozen pre-trained one and a fine-tuned one), where variational score distillation is performed in latent space to ensure that the output of $\epsilon_\theta$ follows HQ natural image distribution. The regularization loss will be back-propagated to update $E_\theta$ and $\epsilon_\theta$. Once training is finished, only $E_\theta$, $\epsilon_\theta$ and $D_\theta$ will be used in inference.

output, $\mathcal{L}_{reg}$ is the regularization term to exploit the prior information of natural images, and scalar $\lambda$ is the balance parameter. Many conventional ISR methods [13, 29, 65] restore the desired HQ image by assuming simple and known degradation models and employing hand-crafted natural image prior models (*i.e.*, image sparsity based prior [54]).

However, the performance of such optimization-based methods is largely hindered by two factors. First, the degradation function $\Phi$ is often unknown and hard to model in real-world scenarios. Second, the hand-crafted regularization terms $\mathcal{L}_{reg}$ are hard to effectively model the complex natural image priors. With the development of deep-learning techniques, it has become prevalent to learn a neural network $G_\theta$, which is parameterized by $\theta$, from a training dataset $S$ of $(\boldsymbol{x}_L, \boldsymbol{x}_H)$ pairs to map the LQ image to an HQ image. The network training can be described as the following learning problem:

$$\theta^* = \operatorname{argmin}_\theta \mathbb{E}_{(\boldsymbol{x}_L, \boldsymbol{x}_H) \sim S} \left[ \mathcal{L}_{\text{data}} \left( G_\theta(\boldsymbol{x}_L), \boldsymbol{x}_H \right) + \lambda \mathcal{L}_{\text{reg}} \left( G_\theta(\boldsymbol{x}_L) \right) \right], \tag{1}$$

where $\mathcal{L}_{\text{data}}$ and $\mathcal{L}_{\text{reg}}$ are the loss functions. $\mathcal{L}_{\text{data}}$ enforces that the network output $\hat{\boldsymbol{x}}_H = G_\theta(\boldsymbol{x}_L)$ can approach to the ground-truth $\boldsymbol{x}_H$ as much as possible, which can be quantified by metrics such as $L_1$ norm, $L_2$ norm and LPIPS [64]. Using only the $\mathcal{L}_{\text{data}}$ loss to train the network $G_\theta$ from scratch may over-fit the training dataset. In this work, we propose to finetune a pre-trained generative network, more specifically the SD [36] network, to improve the generalization capability of $G_\theta$. In addition, the regularization loss $\mathcal{L}_{\text{reg}}$ is critical to improve the generalization capability of $G_\theta$, as well as enhance the naturalness of output HQ images $\hat{\boldsymbol{x}}_H$. Suppose that we have the distribution of real-world HQ images, denoted by $p(\boldsymbol{x}_H)$, the KL divergence [8] is an ideal choice to serve as the loss function of $\mathcal{L}_{\text{reg}}$; that is, the distribution of restored HQ images, denoted by $q_\theta(\hat{\boldsymbol{x}}_H)$, should be identical to $p(\boldsymbol{x}_H)$ as much as possible. The regularization loss can be defined as:

$$\mathcal{L}_{\text{reg}} = \mathcal{D}_{\text{KL}} \left( q_\theta(\hat{\boldsymbol{x}}_H) \| p(\boldsymbol{x}_H) \right). \tag{2}$$

Existing works [24, 46] mostly instantiate the above objective via adversarial training [14], which involves learning a discriminator to differentiate between the generated HQ image $\hat{\boldsymbol{x}}_H$ and the real HQ image $\boldsymbol{x}_H$, and updating the generator $G_\theta$ to make $\hat{\boldsymbol{x}}_H$ and $\boldsymbol{x}_H$ indistinguishable. However, the discriminators are often trained from scratch alongside the generator. They may not be able to acquire the full distribution of HQ images and lack enough discriminative power, resulting in sub-optimal Real-ISR performance.

The recently developed T2I diffusion models such as SD [36] offer new options for us to formulate the loss $\mathcal{L}_{\mathrm{reg}}$. These models, trained on billions of image-text pairs, can effectively depict the natural image distribution in latent space. Some score distillation methods have been reported to employ SD to optimize images by using the KL-divergence as the objective [49, 25, 43]. In particular, variational score distillation (VSD) [49, 58, 10] induces such a KL-divergence based objective from particle-based variational optimization to align the distributions represented by two diffusion models. Based on the above discussions, we propose to instantiate the learning objective in Eq. (1) by designing an efficient and effective one-step diffusion network. In specific, we finetune the pre-trained SD with LoRA [17] as our Real-ISR backbone network and employ VSD as our regularizer to align the distribution of network outputs with natural HQ images. The details are provided in the next section.

## 3.2 One-Step Effective Diffusion Network

**Framework Overview.** As discussed in Sec. 1, the existing SD-based Real-ISR methods [42, 57, 31, 52, 40] perform multiple timesteps to estimate the HQ image with random noise as the starting point and the LQ image as control signal. These approaches are resource-intensive and will inherently introduce randomness. Based on our formulation in Sec. 3.1, we propose a one-step effective diffusion (OSEDiff) network for Real-ISR, whose training framework is shown in Fig. 2. Our generator $G_\theta$ to be trained is composed of a trainable encoder $E_\theta$, a finetuned diffusion network $\epsilon_\theta$ and a frozen decoder $D_\theta$. To ensure the generalization capability of $G_\theta$, the output of the diffusion network $\epsilon_\theta$ will be sent to two regularizer networks, where VSD loss is performed in latent space. The regularization loss are back-propagated to update $E_\theta$ and $\epsilon_\theta$. Once training is finished, only the generator $G_\theta$ will be used in inference. In the following, we will delve into the detailed architecture design of OSEDiff, as well as its associated training losses.

**Network Architecture Design**. Let's denote by $E_\phi$, $\epsilon_\phi$ and $D_\phi$ the VAE encoder, latent diffusion network and VAE decoder of a pretrained SD model, where $\phi$ denotes the model parameters. Inspired by the recent success of LoRA [17] in finetuning SD to downstream tasks [34, 35], we adopt LoRA to fine-tune the pre-trained SD in the Real-ISR task to obtain the desired generator $G_\theta$.

As shown in the left part of Fig. 2, to maintain SD's original generation capacity, we introduce trainable LoRA [17] layers to the encoder $E_\phi$ and the diffusion network $\epsilon_\phi$, finetuning them into $E_\theta$ and $\epsilon_\theta$ with our training data. For the decoder, we fix its parameters and directly set $D_\theta = D_\phi$. This is to ensure that the output space of the diffusion network remains consistent with the regularizers.

Recall that the diffusion model diffuses the input latent feature $z$ through $z_t = \alpha_t z + \beta_t \epsilon$, where $\alpha_t, \beta_t$ are scalars that are dependent to diffusion timestep $t \in \{1, \cdots, T\}$ [16]. With a neural network that can predict the noise in $z_t$, denoted as $\hat{\epsilon}$, the denoised latent can be obtained as $\hat{z}_0 = \frac{z_t - \beta_t \hat{\epsilon}}{\alpha_t}$, which is expected to be cleaner and more photo-realistic than $z_t$. Moreover, SD is a text-conditioned generation model. By extracting the text embeddings [36], denoted by $c_y$, from the given text description $y$, the noise prediction can be performed as $\hat{\epsilon} = \epsilon_\theta(z_t; t, c_y)$.

We adapt the above text-to-image denoising process to the Real-ISR task, and formulate the LQ-to-HQ latent transformation $F_\theta$ as a text-conditioned image-to-image denoising process as:

$$\hat{z}_H = F_\theta(z_L; c_y) \triangleq \frac{z_L - \beta_T \epsilon_\theta(z_L; T, c_y)}{\alpha_T}, \tag{3}$$

where we conduct only one-step denoising on the LQ latent $z_L$, without introducing any noise, at the $T$-th diffusion timestep. The denoising output $\hat{z}_H$ is expected be more photo-realistic than $z_L$. As for the text embeddings, we apply some text prompt extractor, such as the DAPE [52], to LQ input $x_L$, and obtain $c_y = Y(x_L)$. Finally, the whole LQ-to-HQ image synthesis can be written as:

$$\hat{x}_H = G_\theta(x_L) \triangleq D_\theta(F_\theta(E_\theta(x_L); Y(x_L))). \tag{4}$$

As mentioned in Sec. 3.1, to improve the performance for a Real-ISR model, it is required to supervise the generator training with both the data term $\mathcal{L}_{\mathrm{data}}$ and regularization term $\mathcal{L}_{\mathrm{reg}}$. As shown in the right part of Fig. 2, we propose to adapt VSD [49] as the regularization term. Apart from utilizing the SD model as the pre-trained regularizer $\epsilon_\phi$, VSD also introduces a finetuned regularizer, *i.e.*, a latent diffusion module finetuned on the distribution $q_\theta(\hat{x}_H)$ of generated images with LoRA. We denote this finetuned diffusion module as $\epsilon_{\phi'}$.

**Training Loss**. We train the generator $G_\theta$ with the data loss $\mathcal{L}_{\text{data}}$ and regularization loss $\mathcal{L}_{\text{reg}}$. We set $\mathcal{L}_{\text{data}}$ as the weighted sum of MSE loss and LPIPS loss:

$$\mathcal{L}_{\text{data}}\left(G_\theta(\boldsymbol{x}_L), \boldsymbol{x}_H\right) = \mathcal{L}_{\text{MSE}}\left(G_\theta(\boldsymbol{x}_L), \boldsymbol{x}_H\right) + \lambda_1 \mathcal{L}_{\text{LPIPS}}\left(G_\theta(\boldsymbol{x}_L), \boldsymbol{x}_H\right), \tag{5}$$

where $\lambda_1$ is a weighting scalar. As for $\mathcal{L}_{\text{reg}}$, we adopt the VSD loss via:

$$\mathcal{L}_{\text{reg}}\left(G_\theta(\boldsymbol{x}_L)\right) = \mathcal{L}_{\text{VSD}}\left(G_\theta(\boldsymbol{x}_L), c_y\right) = \mathcal{L}_{\text{VSD}}\left(G_\theta(\boldsymbol{x}_L), Y(\boldsymbol{x}_L)\right). \tag{6}$$

Given any trainable image-shape feature $\boldsymbol{x}$, its latent code $\boldsymbol{z} = E_\phi(\boldsymbol{x})$ and encoded text prompt condition $c_y$, VSD optimizes $\boldsymbol{x}$ to make it consistent with the text prompt $y$ via:

$$\nabla_{\boldsymbol{x}} \mathcal{L}_{\text{VSD}}\left(\boldsymbol{x}, c_y\right) = \mathbb{E}_{t,\epsilon}\left[\omega(t)\left(\boldsymbol{\epsilon}_\phi(\boldsymbol{z}_t; t, c_y) - \boldsymbol{\epsilon}_{\phi'}(\boldsymbol{z}_t; t, c_y)\right)\frac{\partial \boldsymbol{z}}{\partial \boldsymbol{x}}\right], \tag{7}$$

where the expectation of the gradient is conducted over all diffusion timesteps $t \in \{1, \cdots, T\}$ and $\epsilon \sim \mathcal{N}(0, \boldsymbol{I})$. Therefore, the overall training objective for the generator $G_\theta$ is:

$$\mathcal{L}\left(G_\theta(\boldsymbol{x}_L), \boldsymbol{x}_H\right) = \mathcal{L}_{\text{data}}\left(G_\theta(\boldsymbol{x}_L), \boldsymbol{x}_H\right) + \lambda_2 \mathcal{L}_{\text{reg}}\left(G_\theta(\boldsymbol{x}_L)\right), \tag{8}$$

where $\lambda_2$ is a weighting scalar. Besides, as required by VSD, the finetuned regularizer $\boldsymbol{\epsilon}_{\phi'}$ should also be trainable, and its training objective is:

$$\mathcal{L}_{\text{diff}} = \mathbb{E}_{t,\epsilon,c_y=Y(\boldsymbol{x}_L),\hat{\boldsymbol{z}}_H=F_\theta(E_\theta(\boldsymbol{x}_L);Y(\boldsymbol{x}_L))} \mathcal{L}_{\text{MSE}}\left(\boldsymbol{\epsilon}_{\phi'}\left(\alpha_t \hat{\boldsymbol{z}}_H + \beta_t \epsilon; t, c_y\right), \epsilon\right). \tag{9}$$

Note that the above $\mathcal{L}_{\text{diff}}$ loss is only applied to update $\boldsymbol{\epsilon}_{\phi'}$. The whole algorithm to illustrate the training pipeline can be found in the **Appendix**.

**VSD in Latent Space**. The original VSD computes the gradients in the image space. When it is used to train an SD-based generator network, there will be repetitive latent decoding/encoding in computing $\mathcal{L}_{\text{reg}}$. This is costly and makes the regularization less effective. Considering the fact that a well-trained VAE should satisfy $E_\phi(\boldsymbol{x}) = E_\phi(D_\phi(\boldsymbol{z})) \approx \boldsymbol{z}$, we can approximately let $E_\phi(\hat{\boldsymbol{x}}_H) = \hat{\boldsymbol{z}}_H$. In this case, we can eliminate the redundant latent encoding/decoding in computing the regularization loss, as we follow DMD [58] to optimize the distribution loss in the latent state space rather than in the noise space. The gradient of the regularization loss w.r.t. the network parameter $\theta$ in the latent space is:

$$\begin{aligned}
\nabla_\theta \mathcal{L}_{\text{VSD}}(G_\theta(\boldsymbol{x}_L), c_y) &= \nabla_{\hat{\boldsymbol{x}}_H} \mathcal{L}_{\text{VSD}}(\hat{\boldsymbol{x}}_H, c_y)\frac{\partial \hat{\boldsymbol{x}}_H}{\partial \theta} \\
&= \mathbb{E}_{t,\epsilon,\hat{\boldsymbol{z}}_t=\alpha_t E_\phi(\hat{\boldsymbol{x}}_H)+\beta_t \epsilon}\left[\omega(t)\left(\boldsymbol{\epsilon}_\phi(\hat{\boldsymbol{z}}_t; t, c_y) - \boldsymbol{\epsilon}_{\phi'}(\hat{\boldsymbol{z}}_t; t, c_y)\right)\frac{\partial \hat{\boldsymbol{z}}_H}{\partial \hat{\boldsymbol{x}}_H}\frac{\partial \hat{\boldsymbol{x}}_H}{\partial \theta}\right] \\
&= \mathbb{E}_{t,\epsilon,\hat{\boldsymbol{z}}_t=\alpha_t \hat{\boldsymbol{z}}_H+\beta_t \epsilon}\left[\omega(t)\left(\boldsymbol{\epsilon}_\phi(\hat{\boldsymbol{z}}_t; t, c_y) - \boldsymbol{\epsilon}_{\phi'}(\hat{\boldsymbol{z}}_t; t, c_y)\right)\frac{\partial \hat{\boldsymbol{z}}_H}{\partial \theta}\right].
\end{aligned} \tag{10}$$

## 4 Experiments

### 4.1 Experimental Settings

**Training and Testing Datasets.** Prior works [42, 57, 31, 52] employed different training datasets, making unified training standards for Real-ISR difficult to establish. For simplicity, we adopt SeeSR's setup [52] and train OSEDiff using the LSDIR [26] dataset and the first 10K face images from FFHQ [19]. The degradation pipeline of Real-ESRGAN [45] is used to synthesize LQ-HQ training pairs. We evaluate OSEDiff and compare it with competing methods using the test set provided by StableSR [42], including both synthetic and real-world data. The synthetic data includes 3000 images of size $512 \times 512$, whose GT are randomly cropped from DIV2K-Val [2] and degraded using the Real-ESRGAN pipeline [45]. The real-world data include LQ-HQ pairs from RealSR [3] and DRealSR [51], with sizes of $128 \times 128$ and $512 \times 512$, respectively.

**Compared Methods.** We compare OSEDiff with state-of-the-art DM-based Real-ISR methods, including StableSR [42], ResShift [60], PASD [57], DiffBIR [31], SeeSR [52] and SinSR [48]. Among them, StableSR, PASD, DiffBIR, and SeeSR are all based on the pre-trained SD model. ResShift trains a DM from scratch in the pixel domain, while SinSR is a one-step model distilled

Table 1: Quantitative comparison with state-of-the-art methods on both synthetic and real-world benchmarks. 's' denotes the number of diffusion reverse steps in the method. The best and second best results of each metric are highlighted in **red** and **blue**, respectively.

| Datasets | Methods | PSNR↑ | SSIM↑ | LPIPS↓ | DISTS↓ | FID↓ | NIQE↓ | MUSIQ↑ | MANIQA↑ | CLIPIQA↑ |
|---|---|---|---|---|---|---|---|---|---|---|
| DIV2K-Val | StableSR-s200 | 23.26 | 0.5726 | **0.3113** | 0.2048 | **24.44** | 4.7581 | 65.92 | 0.6192 | 0.6771 |
| | DiffBIR-s50 | 23.64 | 0.5647 | 0.3524 | 0.2128 | 30.72 | **4.7042** | 65.81 | 0.6210 | 0.6704 |
| | SeeSR-s50 | 23.68 | 0.6043 | 0.3194 | **0.1968** | **25.90** | 4.8102 | **68.67** | **0.6240** | **0.6936** |
| | PASD-s20 | 23.14 | 0.5505 | 0.3571 | 0.2207 | 29.20 | **4.3617** | **68.95** | **0.6483** | **0.6788** |
| | ResShift-s15 | **24.65** | **0.6181** | 0.3349 | 0.2213 | 36.11 | 6.8212 | 61.09 | 0.5454 | 0.6071 |
| | SinSR-s1 | **24.41** | 0.6018 | 0.3240 | 0.2066 | 35.57 | 6.0159 | 62.82 | 0.5386 | 0.6471 |
| | OSEDiff-s1 | 23.72 | **0.6108** | **0.2941** | **0.1976** | 26.32 | 4.7097 | 67.97 | 0.6148 | 0.6683 |
| DrealSR | StableSR-s200 | 28.03 | 0.7536 | 0.3284 | **0.2269** | 148.98 | 6.5239 | 58.51 | 0.5601 | 0.6356 |
| | DiffBIR-s50 | 26.71 | 0.6571 | 0.4557 | 0.2748 | 166.79 | **6.3124** | 61.07 | 0.5930 | 0.6395 |
| | SeeSR-s50 | 28.17 | **0.7691** | **0.3189** | 0.2315 | **147.39** | 6.3967 | **64.93** | **0.6042** | 0.6804 |
| | PASD-s20 | 27.36 | 0.7073 | 0.3760 | 0.2531 | 156.13 | **5.5474** | 64.87 | **0.6169** | **0.6808** |
| | ResShift-s15 | **28.46** | 0.7673 | 0.4006 | 0.2656 | 172.26 | 8.1249 | 50.60 | 0.4586 | 0.5342 |
| | SinSR-s1 | **28.36** | 0.7515 | 0.3665 | 0.2485 | 170.57 | 6.9907 | 55.33 | 0.4884 | 0.6383 |
| | OSEDiff-s1 | 27.92 | **0.7835** | **0.2968** | **0.2165** | **135.30** | 6.4902 | 64.65 | 0.5899 | **0.6963** |
| RealSR | StableSR-s200 | 24.70 | 0.7085 | 0.3018 | 0.2288 | 128.51 | 5.9122 | 65.78 | 0.6221 | 0.6178 |
| | DiffBIR-s50 | 24.75 | 0.6567 | 0.3636 | 0.2312 | 128.99 | 5.5346 | 64.98 | 0.6246 | 0.6463 |
| | SeeSR-s50 | 25.18 | 0.7216 | **0.3009** | **0.2223** | 125.55 | **5.4081** | **69.77** | **0.6442** | 0.6612 |
| | PASD-s20 | 25.21 | 0.6798 | 0.3380 | 0.2260 | **124.29** | **5.4137** | 68.75 | **0.6487** | **0.6620** |
| | ResShift-s15 | **26.31** | **0.7421** | 0.3460 | 0.2498 | 141.71 | 7.2635 | 58.43 | 0.5285 | 0.5444 |
| | SinSR-s1 | **26.28** | **0.7347** | 0.3188 | 0.2353 | 135.93 | 6.2872 | 60.80 | 0.5385 | 0.6122 |
| | OSEDiff-s1 | 25.15 | 0.7341 | **0.2921** | **0.2128** | **123.49** | 5.6476 | **69.09** | 0.6326 | **0.6693** |

Table 2: Complexity comparison among different methods. All methods are tested with an input image of size $512 \times 512$, and the inference time is measured on an A100 GPU.

| | StableSR | DiffBIR | SeeSR | PASD | ResShift | SinSR | OSEDiff |
|---|---|---|---|---|---|---|---|
| Inference Step | 200 | 50 | 50 | 20 | 15 | 1 | 1 |
| Inference Time (s) | 11.50 | 2.72 | 4.30 | 2.80 | 0.71 | 0.13 | 0.11 |
| MACs (G) | 79940 | 24234 | 65857 | 29125 | 5491 | 2649 | 2265 |
| # Total Param (M) | 1410 | 1717 | 2524 | 1900 | 119 | 119 | 1775 |
| # Trainable Param (M) | 150.0 | 380.0 | 749.9 | 625.0 | 118.6 | 118.6 | 8.5 |

from ResShift. Note that we do not compare with the recent method SUPIR [59] because it tends to generate rich yet excessive details, which are however unfaithful to the input image.

For those GAN-based Real-ISR methods, including BSRGAN [61], Real-ESRGAN [45], LDL [27], and FeMaSR [4], we put their results into the **Appendix**.

**Evaluation Metrics.** To provide a comprehensive and holistic assessment on the performance of different methods, we employ a range of full-reference and no-reference metrics. PSNR and SSIM [50] (calculated on the Y channel in YCbCr space) are reference-based fidelity measures, while LPIPS [64], DISTS [12] are reference-based perceptual quality measures. FID [15] evaluates the distance of distributions between GT and restored images. NIQE [62], MANIQA-pipal [55], MUSIQ [22], and CLIPIQA [41] are no-reference image quality measures. We also conduct a user study, which is presented in the **Appendix**.

**Implementation Details.** We train OSEDiff with the AdamW optimizer [33] at a learning rate of 5e-5. The entire training process took approximately 1 day on 4 NVIDIA A100 GPUs with a batch size of 16. The rank of LoRA in the VAE Encoder, diffusion network, and finetuned regularizer is set to 4. For the text prompt extractor, although advanced multimodal language models [32] can provide detailed text descriptions, they come at a high inference cost. We adopt the degradation-aware prompt extraction (DAPE) module in SeeSR [52] to extract text prompts. The SD 2.1-base is used as the pre-trained T2I model. The weighting scalars $\lambda_1$ and $\lambda_2$ are set to 2 and 1, respectively.

### 4.2 Comparison with State-of-the-Arts

**Quantitative Comparisons.** The quantitative comparisons among the competing methods on the three datasets are presented in Table 1. We can have the following observations. (1) First, OSEDiff exhibits clear advantages over competing methods in full-reference perceptual quality metrics LPIPS and DISTS, distribution alignment metric FID, and semantic quality metric CLIPIQA, especially

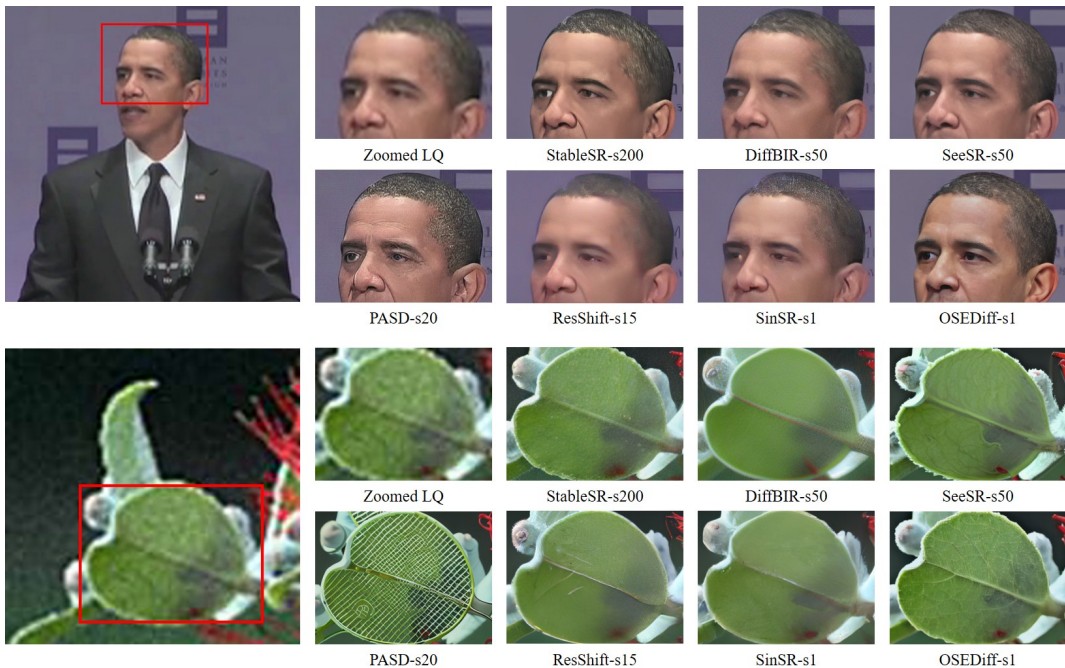

Figure 3: Qualitative comparisons of different Real-ISR methods. Please zoom in for a better view.

on the two real-world datasets DrealSR and RealSR. (2) Second, SeeSR and PASD show better no-reference metrics like NIQE, MUSIQ and MANIQA. This is because these multi-step methods can produce rich image details in the diffusion process, which are preferred by no-reference metrics. (3) Third, ResShift and its distilled version SinSR show better full-reference fidelity metrics such as PSNR. This is mainly because they train a DM from scratch specifically for the restoration purpose, instead of exploring the pre-trained T2I model such as SD. However, ResShift and SinSR show poor perceptual quality metrics than other methods.

**Qualitative Comparisons.** Fig. 3 presents visual comparisons of different Real-ISR methods. As illustrated in the first example, ResShift and SinSR severely blur the facial details due to the lack of pre-trained image priors. StableSR, DiffBIR and SeeSR reconstruct more facial details by exploring the image prior in pre-trained SD model. PASD generates excessive yet unnatural details. Though OSEDiff performs only one step forward propagation, it reproduces realistic and superior facial details to other methods. Similar conclusion can be drawn from the second example. StableSR and DiffBIR are limited in generating rich textures due to the lack of text prompts. PASD suffers from incorrect semantic generation because its prompt extraction module is not robust to degradation. While SeeSR utilizes degradation-aware semantic cues to stimulate image generation priors, the generated leaf veins are unnatural, which may be influenced by its random noise sampling. In contrast, OSEDiff can generate detailed and natural leaf veins. More visualization comparisons and the results of subjective user study can be found in the **Appendix**.

**Complexity Comparisons.** We further compare the complexity of competing DM-based Real-ISR models in Table 2, including the number of inference steps, inference time, and trainable parameters. All methods are tested on an A100 GPU with an input image of size $512 \times 512$. OSEDiff has the fewest trainable parameters, and the trained LoRA layers can be merged into the original SD to further reduce the computational cost. By using only one forward pass, OSEDiff has significant advantage in inference time over multi-step methods. Specifically, OSEDiff demonstrates a substantial speed advantage, being approximately 105 times faster than StableSR, 39 times faster than SeeSR, and 6 times faster than ResShift. When compared to the single-step method SinSR, OSEDiff not only achieves faster inference but also delivers significantly higher output quality. In terms of complexity, OSEDiff requires the lowest MACs at just 2265G, as it operates with only a single diffusion step. In contrast, methods like StableSR, which require 200 steps, incur substantially higher MACs (*e.g.*, 79940G). Regarding trainable parameters, OSEDiff is highly parameter-efficient, requiring only 8.5M parameters (LoRA layers), compared to models such as SeeSR, which necessitates 749.9M parameters. This highlights the efficiency of OSEDiff during the training process.

Table 3: Comparison of different losses on the RealSR benchmark.

| | PSNR↑ | LPIPS↓ | MUSIQ↑ | CLIPIQA↑ |
|---|---|---|---|---|
| w/o VSD loss | 25.42 | 0.2934 | 65.23 | 0.5876 |
| GAN loss | 25.00 | 0.2760 | 67.29 | 0.6254 |
| VSD loss in image domain | 25.05 | 0.2759 | 67.90 | 0.6256 |
| OSEDiff | 25.15 | 0.2921 | 69.09 | 0.6693 |

Table 4: Comparison of different text prompt extractors on the DrealSR benchmark.

| Prompt Extraction Methods | PSNR↑ | SSIM↑ | LPIPS↓ | DISTS↓ | FID↓ | NIQE↓ | MUSIQ↑ | MANIQA↑ | CLIPIQA↑ | Prompt Extraction Time (s) |
|---|---|---|---|---|---|---|---|---|---|---|
| Null | 28.51 | 0.7910 | 0.2896 | 0.2080 | 126.59 | 6.6436 | 62.13 | 0.5782 | 0.6599 | 0 |
| DAPE | 27.92 | 0.7835 | 0.2968 | 0.2165 | 135.30 | 6.4902 | 64.65 | 0.5899 | 0.6963 | 0.02 |
| LLaVA-v1.5 | 27.72 | 0.7735 | 0.3109 | 0.2249 | 149.45 | 6.4119 | 65.70 | 0.6038 | 0.7033 | 3.53 |

Table 5: Comparison of LoRA in VAE encoder with different ranks.

| Rank | PSNR↑ | DISTS↓ | MUSIQ↑ | NIQE↓ |
|---|---|---|---|---|
| 2 | - | - | - | - |
| 4 | 25.15 | 0.2128 | 69.09 | 5.6479 |
| 8 | 24.86 | 0.2134 | 68.37 | 5.8184 |

Table 6: Comparison of LoRA in UNet with different ranks.

| Rank | PSNR↑ | DISTS↓ | MUSIQ↑ | NIQE↓ |
|---|---|---|---|---|
| 2 | 25.28 | 0.2154 | 68.89 | 5.7171 |
| 4 | 25.15 | 0.2128 | 69.09 | 5.6479 |
| 8 | 24.87 | 0.2115 | 68.39 | 5.8184 |

Table 7: Ablation studies on finetuning the VAE encoder and decoder on the RealSR benchmark.

| | Train VAE Encoder | Train VAE Decoder | PSNR↑ | DISTS↓ | LPIPS↓ | CLIPIQA↑ | MUSIQ↑ | NIQE↓ |
|---|---|---|---|---|---|---|---|---|
| (1) | × | × | 25.27 | 0.1966 | 0.2656 | 0.5303 | 58.99 | 6.5496 |
| (2) | × | ✓ | 25.30 | 0.2049 | 0.2829 | 0.5604 | 65.83 | 6.6291 |
| (3) | ✓ | ✓ | 25.59 | 0.2141 | 0.3017 | 0.5778 | 65.92 | 6.9845 |
| OSEDiff | ✓ | × | 25.15 | 0.2128 | 0.2921 | 0.6693 | 69.09 | 5.6479 |

## 4.3 Ablation Study

**Effectiveness of VSD Loss.** To validate the effectiveness of our VSD loss in latent space, we perform ablation studies by removing the VSD loss, replacing it with the GAN loss used in [35], and applying VSD loss in the image domain. The results on the RealSR test set are shown in Table 3. We can see that without using the VSD loss, the perceptual quality metrics are significantly degraded because it is hard to ensure good visual quality using only MSE loss and even LPIPS loss [64]. Using GAN loss and VSD loss in the image domain can improve the performance, but the results are not as good as applying VSD loss in the latent domain. Our proposed OSEDiff can effectively align the distribution of Real-ISR outputs by performing VSD regularization in the latent domain.

**Comparison on Text Prompt Extractors.** We then conduct experiments to evaluate the effect of different text prompt extractors on the Real-ISR results. We test three options. The first option does not employ text prompts. The second option uses the DAPE module in SeeSR [52] to extract degradation-aware tag-style prompts, as we used in our main experiments. The third option uses LLaVA-v1.5 [32] to extract long text descriptions after removing the degradation of input LQ images, as used in SUPIR [59]. We retrain the models based on different prompt extraction methods. The ablation results are shown in Table 4.

One can see that without using text prompts as inputs, those full-reference metrics such PSNR, SSIM, LPIPS, DISTS and even FID can improve, while those no-reference metrics such as MUSIQ, MANIQA and CLIPIQA become worse. By using DAPE and LLaVA to extract text prompts, the generation capability of the pre-trained T2I SD model can be triggered, resulting in richer synthesized details, which however can reduce the full-reference indices. A visual example is shown in Figure 4. We see that while LLaVA extracts significantly longer text prompts than DAPE, they produce a similar amount of visual details. However, it is worth mentioning that the MLLM model LLaVA is very costly, increasing the inference time of DAPE by 170 times. Considering the cost-effectiveness, we ultimately choose DAPE as the text prompt extractor in OSEDiff.

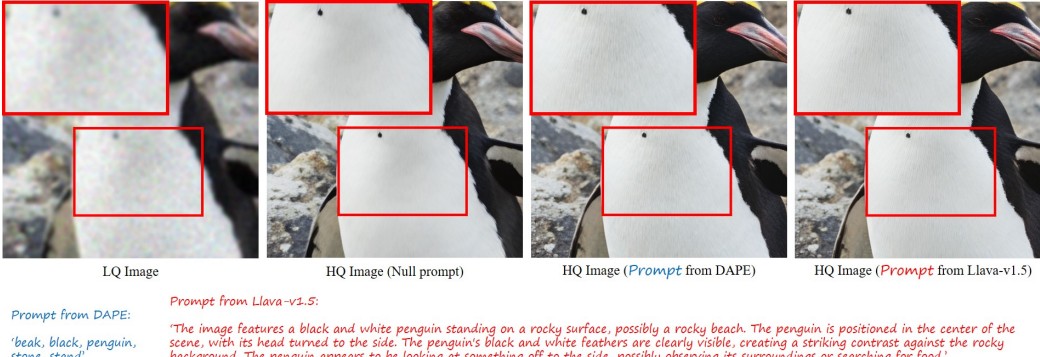

| LQ Image | HQ Image (Null prompt) | HQ Image (*Prompt* from DAPE) | HQ Image (*Prompt* from Llava-v1.5) |

Prompt from DAPE:

'beak, black, penguin, stone, stand'

Prompt from Llava-v1.5:

'The image features a black and white penguin standing on a rocky surface, possibly a rocky beach. The penguin is positioned in the center of the scene, with its head turned to the side. The penguin's black and white feathers are clearly visible, creating a striking contrast against the rocky background. The penguin appears to be looking at something off to the side, possibly observing its surroundings or searching for food.'

Figure 4: The impact of different prompt extraction methods. Please zoom in for a better view.

**Setting of LoRA Rank.** When finetuning the VAE encoder and the UNet, we need to set the rank of LoRA layers. Here we evaluate the effect of different LoRA ranks on the Real-ISR performance by using the RealSR benchmark [3]. The results are shown in Tables 5 and 6, respectively. As shown in Table 5, if a too small LoRA rank, such as 2, is set for the VAE encoder, the training will be unstable and cannot converge. On the other hand, if a higher LoRA rank, such as 8, is used for the VAE encoder, it may overfit in estimating image degradation, losing some image details in the output, as evidenced by the PSNR, DISTS, MUSIQ and NIQE indices. We find that setting the rank to 4 can achieve a balanced result for the VAE encoder. Similar conclusions can be made for the setting of LoRA rank on UNet. As can be seen from Table 6, a rank of 4 strikes a good balance. Therefore, we set the rank as 4 for both the VAE encoder and UNet LoRA layers.

**The Finetuning on the VAE Encoder and Decoder.** We conducted ablation studies to examine the impact of finetuning the VAE encoder and decoder, as shown Table 7. In the first row, where neither the VAE encoder nor decoder is finetuned, the results show poor perception performance. Comparing with OSEDiff, where only the VAE encoder is finetuned, we observe significant improvements in perceptual quality (*e.g.*, MUSIQ improves from 58.99 to 69.09). This demonstrates that finetuning the VAE encoder is important for removing degradation and enhancing overall performance. When comparing the third row, where both the VAE encoder and decoder are finetuned, with OSEDiff, where only the encoder is trained and the decoder is fixed, we note that OSEDiff also achieves better perceptual quality (CLIPIQ improves from 0.5778 to 0.6693). This indicates that fixing the VAE decoder is important to ensure that the UNet output remains in the original VAE latent space, which helps minimizing the VSD loss more effectively. Thus, finetuning the VAE encoder is important to remove degradation, while fixing the VAE decoder helps maintaining stability in the latent space, leading to better perceptual quality.

## 5   Conclusion and Limitation

We proposed OSEDiff, a one-step effective diffusion network for Real-ISR, by utilizing the pre-trained text-to-image model as both the generator and the regularizer in training. Unlike traditional multi-step diffusion models, OSEDiff directly took the given LQ image as the starting point for diffusion, eliminating the uncertainty associated with random noise. By fine-tuning the pre-trained diffusion network with trainable LoRA layers, OSEDiff can well adapt to the complex real-world image degradations. Meanwhile, we performed the variational score distillation in the latent space to ensure that the model's predicted scores align with those of multi-step pre-trained models, enabling OSEDiff to efficiently produce HQ images in one diffusion step. Our experiments showed that OSEDiff achieved comparable or superior Real-ISR outcomes to previous multi-step diffusion-based methods in both objective metrics and subjective assessments. We believe our exploration can facilitate the practical application of pre-trained T2I models to Real-ISR tasks.

There are some limitations of OSEDiff. First, the details generation capability of OSEDiff can be further improved. Second, like other SD-based methods, OSEDiff is limited in reconstructing fine-scale structures such as small scene texts. We will investigate these problems in further work.

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

# A  Appendix

In the appendix, we provide the following materials:

- Comparison with GAN-based methods (referring to Section 4.1 in the main paper).
- Results of user study (referring to Section 4.1 in the main paper).
- More real-world visual comparisons under scaling factor $4\times$ (referring to Section 4.2 in the main paper).
- Training algorithm of OSEDiff (referring to Section 3.2 in the main paper).

## A.1  Comparison with GAN-based Methods

We compare OSEDiff with four representative GAN-based Real-ISR methods, including BSRGAN [61], Real-ESRGAN [45], LDL [27] and FeMaSR [4]. The results are shown in Table 8. It is not a surprise that GAN-based methods have better fidelity measures such as PSNR and SSIM than OSEDiff. However, OSEDiff has much better perceptual qualify metrics. We also provide visual comparisons in Figure 5. Compared to GAN-based methods, OSEDiff is able to generate realistic and reasonable details, such as squirrel hair, textures of petals, buildings, and leaves.

## A.2  User Study

To further validate the effectiveness of our proposed OSEDiff method, we conducted a user study by using 20 real-world LQ images. An LQ image and its HQ counterparts generated by different Real-ISR methods were presented to volunteers, who were asked to select the best HQ result. The volunteers were instructed to consider two factors when making their decisions: the image perceptual quality and and its content (including structure and texture) consistency with the LQ input, with each factor contributing equally to the final selection.

We randomly selected 20 real-world LQ images from the RealLR200 dataset [52]. Figure 6(a) shows the thumbnails used in the user study, cropped into squares for a convenient layout. We generated the HQ outputs of them by using the DM-based Real-ISR methods StableSR [42], DiffBIR [31], SeeSR [52], PASD [57], ResShift [60], SinSR [48], and OSEDiff. A number of 15 volunteers were invited to participate in the evaluation. The results are shown in Figure 6(b). We see that OSEDiff ranks the second, just lagging slightly behind SeeSR. However, it should be noted that OSEDiff runs over 10 times faster than SeeSR by performing only one-step diffusion.

## A.3  More Visual Comparisons

Figure 7 provides more visual comparisons between OSEDiff and other DM-based methods. One can see that OSEDiff achieves comparable to or even better results than the multi-step diffusion methods in scenarios such as portraits, flower patterns, buildings, animal fur, and letters.

Table 8: Quantitative comparison with GAN-based methods on both synthetic and real-world benchmarks. The best results of each metric are highlighted in **red**.

| Datasets | Methods | PSNR↑ | SSIM↑ | LPIPS↓ | DISTS↓ | FID↓ | NIQE↓ | MUSIQ↑ | MANIQA↑ | CLIPIQA↑ |
|----------|---------|-------|-------|--------|--------|------|-------|--------|---------|----------|
| DIV2K-Val | BSRGAN | **24.58** | 0.6269 | 0.3351 | 0.2275 | 44.23 | 4.7518 | 61.20 | 0.5071 | 0.5247 |
| | Real-ESRGAN | 24.29 | **0.6371** | 0.3112 | 0.2141 | 37.64 | 4.6786 | 61.06 | 0.5501 | 0.5277 |
| | LDL | 23.83 | 0.6344 | 0.3256 | 0.2227 | 42.29 | 4.8554 | 60.04 | 0.5350 | 0.5180 |
| | FeMASR | 23.06 | 0.5887 | 0.3126 | 0.2057 | 35.87 | 4.7410 | 60.83 | 0.5074 | 0.5997 |
| | OSEDiff | 23.72 | 0.6108 | **0.2941** | **0.1976** | **26.32** | **4.7097** | **67.97** | **0.6148** | **0.6683** |
| DrealSR | BSRGAN | **28.75** | 0.8031 | 0.2883 | 0.2142 | 155.63 | 6.5192 | 57.14 | 0.4878 | 0.4915 |
| | Real-ESRGAN | 28.64 | 0.8053 | 0.2847 | **0.2089** | 147.62 | 6.6928 | 54.18 | 0.4907 | 0.4422 |
| | LDL | 28.21 | **0.8126** | **0.2815** | 0.2132 | 155.53 | 7.1298 | 53.85 | 0.4914 | 0.4310 |
| | FeMASR | 26.90 | 0.7572 | 0.3169 | 0.2235 | 157.78 | **5.9073** | 53.74 | 0.4420 | 0.5464 |
| | OSEDiff | 27.92 | 0.7835 | 0.2968 | 0.2165 | **135.30** | 6.4902 | **64.66** | **0.5899** | **0.6963** |
| RealSR | BSRGAN | **26.39** | **0.7654** | **0.2670** | 0.2121 | 141.28 | 5.6567 | 63.21 | 0.5399 | 0.5001 |
| | Real-ESRGAN | 25.69 | 0.7616 | 0.2727 | **0.2063** | 135.18 | 5.8295 | 60.18 | 0.5487 | 0.4449 |
| | LDL | 25.28 | 0.7567 | 0.2766 | 0.2121 | 142.71 | 6.0024 | 60.82 | 0.5485 | 0.4477 |
| | FeMASR | 25.07 | 0.7358 | 0.2942 | 0.2288 | 141.05 | 5.7885 | 58.95 | 0.4865 | 0.5270 |
| | OSEDiff | 25.15 | 0.7341 | 0.2921 | 0.2128 | **123.49** | **5.6476** | **69.09** | **0.6326** | **0.6693** |

## A.4 Algorithm of OSEDiff

The pseudo-code of our OSEDiff training algorithm is summarized as **Algorithm 1**. We follow [49, 58] and use classifier-free guidance (cfg) when calculating $z_\phi$. The cfg value is set to 7.5, and the negative prompts we use are: "*painting, oil painting, illustration, drawing, art, sketch, oil painting, cartoon, CG Style, 3D render, unreal engine, blurring, dirty, messy, worst quality, low quality, frames, watermark, signature, jpeg artifacts, deformed, lowres, over-smooth.*"

---

**Algorithm 1** Training Scheme of OSEDiff

---

**Input:** Training dataset $\mathcal{S}$, pretrained SD parameterized by $\phi$ including VAE encoder $E_\phi$, latent diffusion network $E_\phi$ and VAE decoder $\epsilon_\phi$, prompt extractor $Y$, training iteration $N$

1   Initialize $G_\theta$ parameterized by $\theta$, including
     $E_\theta \leftarrow E_\phi$ with trainable LoRA
     $\epsilon_\theta \leftarrow \epsilon_\phi$ with trainable LoRA
     $D_\theta \leftarrow D_\phi$
2   Initialize $\epsilon_\phi \leftarrow \epsilon_\theta$ with trainable LoRA
3   **for** $i \leftarrow 1$ **to** $N$ **do**
4      Sample $\boldsymbol{x}_L, \boldsymbol{x}_H$ from $\mathcal{S}$
     /* Network forward                                                        */
5      $c_y \leftarrow Y(\boldsymbol{x}_L)$
6      $\boldsymbol{z}_L \leftarrow E_\theta(\boldsymbol{x}_L)$
7      $\hat{\boldsymbol{z}}_H \leftarrow F_\theta(\boldsymbol{z}_L; c_y)$
8      $\hat{\boldsymbol{x}}_H \leftarrow D_\theta(\hat{\boldsymbol{z}}_H)$
     /* Compute data term objective                                           */
9      $\nabla_\theta \mathcal{L}_{\text{data}} \leftarrow [\mathcal{L}_{\text{MSE}}(\hat{\boldsymbol{x}}_H, \boldsymbol{x}_H) + \lambda_1 \mathcal{L}_{\text{LPIPS}}(\hat{\boldsymbol{x}}_H, \boldsymbol{x}_H)] \frac{\partial \hat{\boldsymbol{x}}_H}{\partial \theta}$
     /* Compute regularization objective, following DMD[58]        */
10      Sample $\epsilon$ from $\mathcal{N}(0, \boldsymbol{I})$
11      Sample $t$ from $\{20, \cdots, 980\}$
12      $\hat{\boldsymbol{z}}_t \leftarrow \alpha_t \hat{\boldsymbol{z}}_H + \sigma_t \epsilon$
13      $\boldsymbol{z}_\phi \leftarrow \text{stopgrad}(F_\phi(\hat{\boldsymbol{z}}_t; c_y))$
14      $\boldsymbol{z}_{\phi'} \leftarrow \text{stopgrad}(F_{\phi'}(\hat{\boldsymbol{z}}_t; c_y))$
15      $\omega \leftarrow 1/\text{mean}(\|\boldsymbol{z}_\phi - \hat{\boldsymbol{z}}_H\|)$
16      $\nabla_\theta \mathcal{L}_{\text{reg}} \leftarrow [\omega(\boldsymbol{z}_{\phi'} - \boldsymbol{z}_\phi)] \frac{\partial \hat{\boldsymbol{z}}_H}{\partial \theta}$
     /* Compute regularizer funetuning objective                    */
17      Sample $\epsilon$ from $\mathcal{N}(0, \boldsymbol{I})$
18      Sample $t$ from $\{1, \cdots, T\}$
19      $\boldsymbol{z}_t \leftarrow \alpha_t \text{stopgrad}(\hat{\boldsymbol{z}}_H) + \sigma_t \epsilon$
20      $\mathcal{L}_{\text{diff}} \leftarrow \mathcal{L}_{\text{MSE}}(\epsilon_{\phi'}(\boldsymbol{z}_t; t, c_y), \epsilon)$
     /* Network Parameter Update                                         */
21      Update $\theta$ with $\mathcal{L}_{\text{data}} + \lambda_2 \mathcal{L}_{\text{reg}}$
22      Update $\phi'$ with $\mathcal{L}_{\text{diff}}$
23   **end**
     **Output:** Generator $G_\theta$ including VAE encoder $E_\theta$, latent diffusion network $E_\theta$ and VAE decoder $\epsilon_\theta$

---

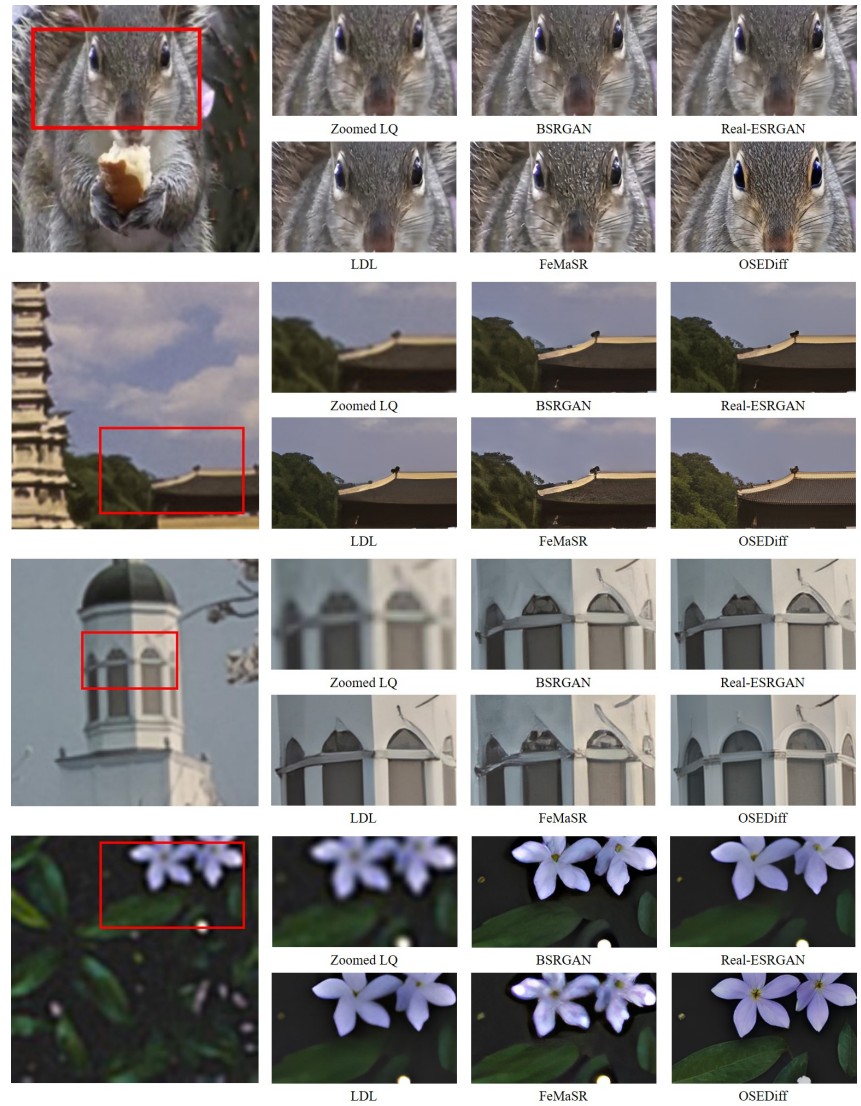

Figure 5: Qualitative comparisons between OSEDiff and GAN-based Real-ISR methods. Please zoom in for a better view.

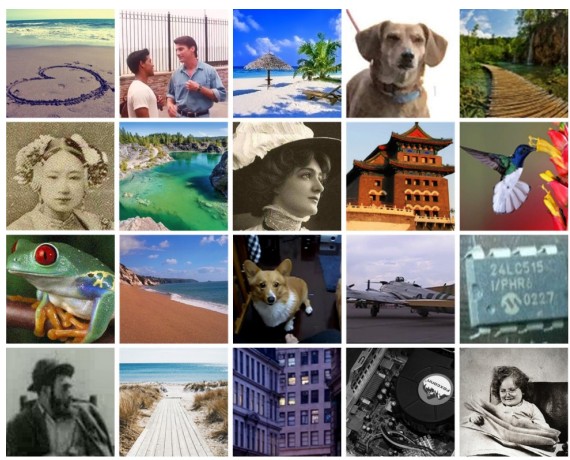

(a) The LQ images used in user study

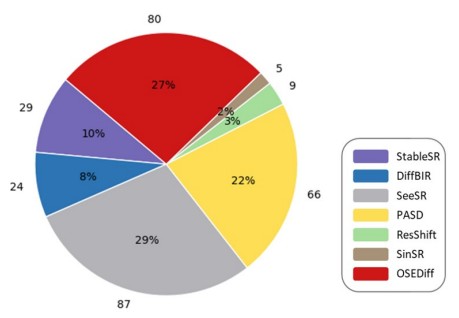

(b) The voting results are from 15 volunteers. The corresponding percentages and numerical counts are displayed with the pie chart.

Figure 6: The LQ images used in user study and the voting results.

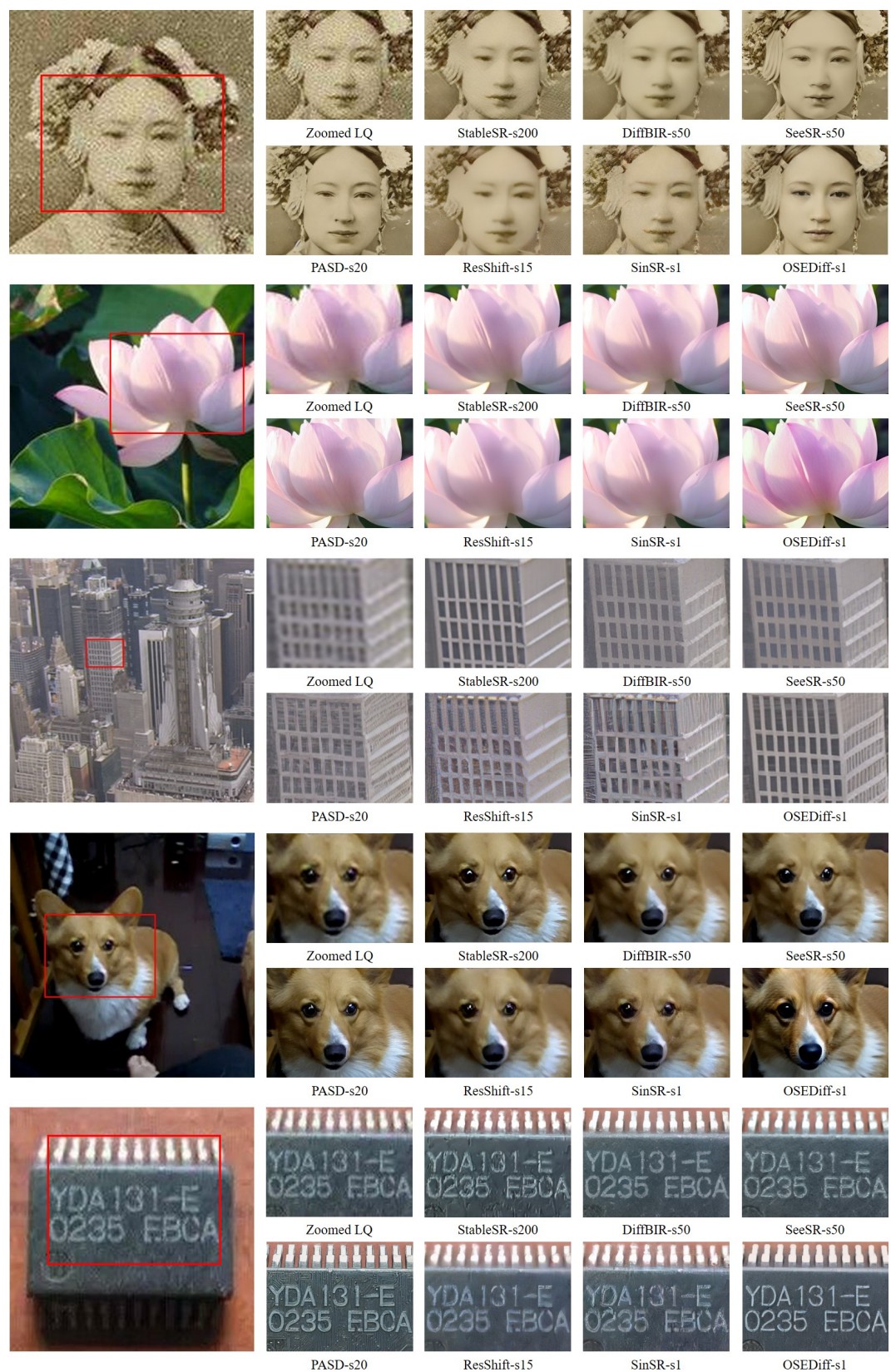

Figure 7: More visulization comparisons of different DM-based Real-ISR methods. Please zoom in for a better view.

