# OpenReview forum: "One-Step Effective Diffusion Network for Real-World  Image Super-Resolution"
_NeurIPS.cc/2024/Conference — NeurIPS 2024 poster_

### Official Review · Reviewer_4vqL · 2024-07-13

**Soundness:** 3
**Presentation:** 3
**Contribution:** 3
**Rating:** 5
**Confidence:** 4

**Summary:**

In this paper, authors propose a novel one-step effective diffusion network, termed as OSEDiff, for the Real-ISR. The proposed OSEDiff adopts LQ image as the input and directly output the final output with the help of the decoder of VAE, thus eliminate the uncertainty introduced by random noise sampling and achieve one-step generation of diffusion model. Besides, the variational score distillation is introduced in the latent space to conduct KL-divergence regularization to enhance the quality of final Real-ISR output. In the experimental part, the proposed method outperforms other baselines in both quantitative and qualitative comparisons.

**Strengths:**

There are several strengths here:
1. One-step diffusion model is attractive and intereting, which will significantly reduce the inference time.
2. The paper is easy to read and understand.
3. The proposed method show comparable results in the experimental part.

**Weaknesses:**

There are several weakness here:
1. Experimental part could be further improved, which will be further demonstrated in the following section.
2. The efficient and params of the proposed method is expected to be detailed demonstrated.

**Questions:**

There are several concerns and suggestions here:
1. I am wondering the comparisons between the proposed method and SinSR. Does the improvement of performance gain from the pre-trained SD?
2. Experiments of the paper is expected to be improved. For instance, total parameters instead of the training parameters are expected to be compared in the Table 2, which is more closely related to the deployment.
3. I am wondering the reason why the setting of "w/o VSD loss" may achieve better PSNR than proposed method. I would be better to provide more visual results of it to demonstrate the effectiveness of the VSD loss.

**Limitations:**

Could be referred to above.

---

> ### Author Rebuttal · Authors · 2024-08-06
>
> **Q1. Comparison with SinSR.**
>
> The improvements of OSEDiff over SinSR mainly come from the pre-trained SD model and the VSD loss. The SD model, pre-trained on large-scale data, contains rich prior knowledge of natural images, enabling effective one-step generation. Additionally, we finetune the multi-step SD model into a single-step network using LoRA and VSD loss. It ensures the one step generation capacity of OSEDiff for restoration tasks while avoiding the lengthy inference time of multiple steps.
>
> **Q2. Total params rather than training params.**
>
> Thanks for the suggestion. The table below show the total number of parameters, FLOPs and the inference time of the competing methods. Please refer to our responses to Q2 of Reviewer KDhM for more discussions. We will add this table and the associated discussions in the revision.
>
> **Table: Complexity comparison among different methods. All methods are tested with an input image of size 512×512 on an A100 GPU.**
>
> |                       | StableSR | DiffBIR | SeeSR | PASD   | ResShift | SinSR | OSEDiff |
> |-----------------------|:--------:|:-------:|:-----:|:------:|:--------:|:-----:|:-------:|
> | **# Total Param (M)** | 1410     | 1717    | 2524  | 1900   | 119      | 119   | 1775    |
> | **FLOPs (G)**         | 79940    | 24234   | 65857 | 29125  | 5491     | 2649  | 2265    |
> | **Inference Step**    | 200      | 50      | 50    | 20     | 15       | 1     | 1       |
> | **Inference Time (s)**| 11.50    | 2.72    | 4.30  | 2.80   | 0.71     | 0.16  | 0.35    |
>
> **Q3. Why is the PSNR lower when using VSD Loss compared to not using it, and more visualization results.**
>
> In the objective function of a general learning-based image restoration framework (refer to Eq. (1) in the main paper), there are two major components: a fidelity term and a regularization term. The fidelity term is usually measured and constrained by $L_2$ or $L_1$ norms, which is friendly to the PSNR metric, while the regularization term is designed based on the employed prior knowledge of natural images. In our work, the VSD loss, which aligns the distribution of network outputs to that of SD prior distribution (i.e., natural image prior distribution), is used as the regularization term. With VSD, the fidelity will be traded-off a little but the perceptual quality will be much enhanced (due to the alignment with natural image prior distribution). Without VSD, the network will focus on optimizing the fidelity term so that the PSNR metric can be improved; however, the perceptual quality will be decreased. Visual examples can be found in Figure 5 of the enclosed PDF file. Without the VSD loss, the PSNR of the restoration results is higher, but they lack semantic details.

---

> > ### Comment · Reviewer_4vqL · 2024-08-13
> >
> > Many thanks for authors' response. Most of the concerns have been satisfied. The only concerns relies on the inference times, I hope the authors could demonstrate more on why the proposed OSEDiff obtains few FLOPs but more inference times than SinSR. It would be helpful for the readers.

---

> ### Author Response · Authors · 2024-08-14
>
> Thank you very much for pointing out this problem. We carefully re-examined OSEDiff's inference code and found that it is a Python decorator function that consumes much time. This function is just to calculate the inference memory usage without affecting anything on the restoration process. We therefore removed this function and reassessed the inference time of different modules for both OSEDiff and SinSR. The results are shown in the table below:
>
> | OSEDiff | DAPE | Text Encoder | VAE Encoder | UNet | VAE Decoder | SUM |
> |:------------:|:------:|:------------:|:-----------:|:------:|:-----------:|:------:|
> | FLOPs (G) | 104 | 22 | 542 | 355 | 1242 | 2265 |
> | Time cost (s)| 0.023 | 0.010 | 0.020 | 0.026 | 0.037 | 0.116 |
>
> | SinSR | VAE Encoder | UNet | VAE Decoder | SUM |
> |:------------:|:-----------:|:------:|:-----------:|:------:|
> | FLOPs (G) | 898 | 202 | 1549 | 2649 |
> | Time cost (s)| 0.034 | 0.045 | 0.051 | 0.130 |
>
> We can see that the overall time consumption of OSEDiff is 0.116s, which is actually less than SinSR (0.130s). Though the DAPE module in OSEDiff costs additional time, the VAE Encoder, UNet, and VAE Decoder modules of OSEDiff consume less time than those of SinSR. Note that although SinSR's UNet has lower FLOPs (202G) than OSEDiff's UNet (355G), its inference time is nearly doubled because SinSR's UNet employs Swin Transformer blocks. The frequent window partitioning operations elevate memory access and data movement costs, resulting in increased latency. Similar observations can be seen for OSEDiff's DAPE module, which also uses a Swin Transformer backbone. Its FLOPs is less than 1/3 of that of UNet, yet its time consumption is comparable to UNet.
>
> The time consumption of SinSR reported in our main paper is 0.16s, which includes the time for converting the model output tensor to an image. We will change it to 0.130s in the revision for fair comparison.
>
> Again, we sincerely thank this reviewer for the careful reading of our paper and indicating this problem. We will correct this issue in the revision.

---

> > ### Comment · Reviewer_4vqL · 2024-08-14
> >
> > Thanks for your response. All my concerns have been addressed. I will keep my positive rating.

---

### Official Review · Reviewer_px3F · 2024-07-13

**Soundness:** 3
**Presentation:** 3
**Contribution:** 2
**Rating:** 5
**Confidence:** 5

**Summary:**

This paper introduces a diffusion-based real-world image super-resolution method, OSEDiff, which can efficiently generate high-quality images in just one diffusion step. Firstly, in order to eliminate the uncertainty introduced by random noise sampling, the authors propose to directly feed the low-quality images without any random noise into the pre-trained SD model which integrates trainable LoRA layers. Furthermore, the variational score distillation for KL-divergence regularization is utilized to align the distribution of generated images with natural image prior, ensuring that the one-step diffusion model could still generate high-quality output. OSEDiff achieves comparable performance to state-of-the-art SD-based real-world image super-resolution methods with fewer trainable parameters and inference time.

**Strengths:**

This paper introduces a one-step effective diffusion network for real-world image super-resolution. The low-quality images without any random noise are utilized as the inputs of the pre-trained SD model and the VSD in latent space are utilized to align the distribution of generated images with natural image prior. The proposed method solves two problems in diffusion-based real-world image super-resolution and it achieves comparable performance to state-of-the-art SD-based real-world image super-resolution methods with fewer trainable parameters and inference time. The figures of this paper are simple and clear, with precise formula expressions, demonstrating a certain clarity.

**Weaknesses:**

(1) The motivation is unclear. In the introduction, the authors use a certain amount of space to mention two major issues in training a Real-ISR model, including the problem of how to build the LQ-HQ training image pairs, but this paper does not propose an innovative solution to this problem.

(2) The explanation of the novelty and its effectiveness are insufficient. 1) Contribution 1, "directly feeding the LQ images into the pre-trained SD model without introducing any random noise," claims to eliminate the uncertainty of output but lacks supporting evidence, raising doubts about its validity. It is recommended to provide visual evidence and analyze why this can eliminate the uncertainty and achieve better Real-ISR performance in detail. 2) Contribution 2, “utilizing variational score distillation for KL-divergence regularization,” it is recommended to provide some visualization of the output of diffusion network which are sent to the two regularizer networks to prove that it can align the distribution of generated images with natural image prior.

(3) The metrics used need more explanation. Since CLIPIQA is text-related, although it has been proved to be effective and generalized in image quality assessment, it is not that convincing to directly apply it to evaluate the quality of real-world super-resolution images. More explanation can be provided.

(4) Details of the network structure are lacking in the figure. In Figure 2, the detailed network structure of VAE encoder and the diffusion network with trainable LoRA layers are lacking. It is recommended to provide detailed network structure.

(5) In “Ablation Study”, the super-resolution performance of the method with text prompt extractor is not competitive enough compared to removing text prompt. Although the qualitative comparisons show that the method with text prompt extractor can generate richer image details, the method without text prompt is much better for several common metrics.

**Questions:**

(1) Could you please provide some visualization of the output of diffusion network which are sent to the two regularizer networks to further prove the effectiveness of the regularization loss in the latent space?

(2) Could you please explain in detail why "directly feeding the LQ images into the pre-trained SD model" can eliminate the uncertainty of output and achieve better Real-ISR performance than the methods which uses inputs with random noise?

(3) Could you please provide detailed network structure of VAE encoder and the diffusion network with trainable LoRA layers?

(4) Could you please provide more qualitative comparisons of the method with/without text prompt extractor to further prove the effectiveness of text prompt or further elaborate on the importance of the text prompt to prove that it is essential?

**Limitations:**

(1) Briefly explain why "directly feeding the LQ images into the pre-trained SD model" can eliminate the uncertainty of output, and why it can achieve better Real-ISR performance than the methods which uses inputs with random noise.

(2) Please provide some visualization of the output of diffusion network which are sent to the two regularizer networks.

---

> ### Author Rebuttal · Authors · 2024-08-06
>
> **Q1. Unclear motivation.**
>
> Thanks for the comments. The goal of this work is to develop an efficient and effective Real-ISR method by using the pre-trained SD prior. In the research and development of Real-ISR methods, how to construct LQ-HQ training pairs is a critical issue. Therefore, we spend a certain amount of space to introduce this problem so that readers can have a more complete understanding of Real-ISR. Based on this reviewer's comments, we will compress this part and make the introduction of LQ-HQ pair more concise.
>
> **Q2. The benefits of reduced uncertainty.**
>
> Thanks for your suggestion. Previous diffusion-based Real-ISR methods apply several denoising steps to generate HQ latents from noisy inputs. However, this process introduces unwanted randomness in the Real-ISR outputs, causing variations with different noise samples. OSEDiff uses LQ image as input to the diffusion network without random noise sampling, enabling a deterministic mapping from LQ to the HQ image. Meanwhile, OSEDiff leverages the VSD loss to enhance the generation capability of the LoRA finetuned network.
>
> As suggested by this reviewer, in Figure 2 of the enclosed PDF file we provide visual comparison of the Real-ISR results of OSEDiff, StableSR and SeeSR. For StableSR and SeeSR, we show their results with two different noise samples. One can see clearly the difference between the two outputs caused by the randomness of noise. For example, the details in StableSR-1 are overly generated, while the result of StableSR-2 is smooth. In contrast, OSEDiff does not involve noise in the input, and it achieves stable Real-ISR result. We will add the visual evidence and analysis in the revision.
>
> **Q3. Visualization of network outputs.**
>
> Thanks for the good suggestion. As suggested by this reviewer, we visualize the distributions of outputs of OSEDiff with and without VSD loss using t-SNE on the RealSR dataset. The distribution of pre-trained SD model is plotted as a reference. We use LLaVa1.5 to generate the captions for the RealSR dataset, and use them as prompts for pre-trained SD, with the inference steps set to 50. As shown in Figure 3 of the enclosed PDF file, the output distribution of OSEDiff with VSD loss is much closer to the pre-trained SD than the OSEDiff without VSD loss. This clearly validates that the VSD loss aligns the  distribution of diffusion network outputs with that of pre-trained SD. We will add this part in the revision.
>
> **Q4. Why use CLIPIQA for Real-ISR evaluation.**
>
> We follow previous works (StableSR, DiffBIR, and SeeSR) to use CLIPIQA as an evaluation metric for fair comparison. The visual examples in Figures 1, 4, and 5 the enclosed PDF file demonstrate that higher CLIPIQA scores generally correspond to better visual quality of the images. However, we agree with this reviewer that highly reliable no-reference IQA remains an unsolved problem, especially for real-world images, and it needs more efforts from the community to find a more robust metric.
>
> **Q5. Details about trainable LoRA module.**
>
> Thanks for the suggestion. We finetuned all layers except the normalization layers using LoRA. We empirically found that fine-tuning the normalization layers causes the model to crash, so we frozen them. We will add more detailed descriptions in the revision.
>
> **Q6. Null prompts vs. Tag prompts.**
>
> Though the results with null prompts score higher on some reference metrics (e.g., PSNR and DISTS) than those with tag-style prompts extracted by DAPE, their visual quality is inferior. This is mainly because the commonly used metrics such as PSNR cannot faithfully reflect the visual quality of images. As shown in Figure 4 of the enclosed PDF file, the results using tag-style prompts have richer leaf vein textures, clearer lines, and less noise. Therefore, we choose tag prompts extracted by DAPE. We will further clarify this in the revision.

---

> > ### Comment · Reviewer_px3F · 2024-08-13
> > **Official Comment by Reviewer px3F**
> >
> > Thanks for the authors' rebuttal. Most of my concerns have been addressed. I still have one question about the SR results in Table 1. We can see from this table that 9 metrics are used for performance evaluation, and part of them are better than other methods. In my opinion, the metrics results of different models of the proposed method may vary significantly. Some models can be chosen to perform well on part of the metrics. In fact, the same applies to the comparison methods, i.e., they can also be tailored to specific metrics like this paper. This paper directly uses the default models of comparison methods, and haven’t chosen models for metrics like this paper. Could this approach potentially make some comparison methods perform better than the proposed method? This can be analyzed in the revised version.

---

> > > ### Author Response · Authors · 2024-08-14
> > >
> > > We are pleased that this reviewer found our responses helpful, and we thank this reviewer for the further question on evaluation metrics, which is actually an open problem for real-world image super-resolution (Real-ISR) evaluation.
> > >
> > > **Q1. Performance evaluation metrics**
> > >
> > > The key contribution of our work is the development of a fast one-step diffusion model for Real-ISR. As can be seen in the main paper, our OSEDiff model significantly reduces the inference time of previous SD-based models (e.g., it is 30x faster than StableSR and 10x faster than SeeSR). Apart from complexity and speed, for the performance evaluation metrics shown in Table 1 of our paper, we actually adopted the ones used in existing works in order for fair comparison with them. We didn’t deliberately select metrics which are advantages to our model.
> > >
> > > Those metrics can be generally classified into two categories: fidelity-oriented ones (e.g., PSNR, SSIM) and perception-oriented ones (e.g., NIQE, MUSIQ, CLIPIQA). For the Real-ISR task, usually we emphasize a little bit more on the perceptual quality of the restoration output. Unfortunately, by far the accurate perceptual image quality assessment (IQA) remains an open problem, and no one single metric can align with human visual perception very well. On the other hand, there are certain conflicts between fidelity-oriented metrics and perception-oriented ones. This also explains why one algorithm performs well on some metrics but not so well on other metrics. If an algorithm is tailored to some metrics, it may become disadvantageous on other metrics.
> > >
> > > Being highly advantageous on inference speed, our approach shows comparable with or even superior results to the competing methods on the nine metrics. The visual comparisons also demonstrate its competitiveness. We will release our codes and trained models so that the peers can test OSEDiff’s performance on different scenarios.
> > >
> > > **Q2. Default models of comparison methods**
> > >
> > > We follow the convention in the community to use the official models of competing methods for comparison, with the parameter settings recommended by the authors. We believe this is the fairest way to compare the different Real-ISR methods. If we tailor one model or change its parameter setting to improve some metrics, other metrics may be deteriorated (see our responses to Q1), and this is unfair for other competing methods.  Nonetheless, we agree that how to better evaluate the performance of Real-ISR models needs the collective efforts from the community.

---

### Official Review · Reviewer_KDhM · 2024-07-13

**Soundness:** 3
**Presentation:** 3
**Contribution:** 3
**Rating:** 7
**Confidence:** 5

**Summary:**

This paper presents a novel approach to real-world image super-resolution for a one-step effective diffusion network (OSEDiff). The proposed OSEDiff effectively eliminates the uncertainty introduced by random noise sampling in previous methods, achieving significant performance improvements across multiple benchmarks, demonstrating its potential for practical applications.

**Strengths:**

1. The focus on one-step diffusion in this paper is particularly noteworthy, as it addresses a critical challenge for the practical application of diffusion models in image super-resolution tasks.
2. The performance gains and visually compelling results suggest the effectiveness of the proposed method for real-world image super-resolution.
3. The paper is well-structured and easy to follow.

**Weaknesses:**

1. The core concept of this paper shares similarities with "One-step Diffusion with Distribution Matching Distillation". A comprehensive comparison with this work, including performance metrics, training efficiency (parameters and FLOPs), and potential limitations, would strengthen the paper's contribution.
2. While Table 2 provides a comparison of trainable parameters, including the total number of model parameters and FLOPs during inference would offer a more complete picture of computational efficiency.
3. There is a lack of discussion on the degradation in LQ images, such as how to ensure that text prompts remain accurate in low-quality scenarios.
4. In Figure 7 (first and second scenarios), some generated details and textures appear to be creatively added rather than strictly reconstructed from the LQ input. Addressing potential limitations in fidelity due to this creative aspect would be valuable.

**Questions:**

No more questions.

**Limitations:**

The authors have discussed about the limitations.

---

> ### Author Rebuttal · Authors · 2024-08-06
>
> **Q1. Comparison with DMD.**
>
> Thanks for the nice suggestion. While both OSEDiff and DMD draw upon the concept of variational distillation from ProlificDreamer, they differ significantly in several aspects. First, DMD is designed for text-to-image tasks, whereas OSEDiff is tailored for image restoration tasks, aiming to reconstruct an HQ image from its LQ counterpart. To achieve this goal, OSEDiff employs a trainable VAE encoder to remove degradation from LQ images, and it directly takes the LQ as input to the diffusion network to avoid the uncertainty caused by random noise sampling. Trainable LoRA layers are introduced to the diffusion UNet to adapt it to the restoration task. In contrast, DMD lacks these specific designs tailored for image restoration tasks. Furthermore, DMD requires full parameter fine-tuning of two SD models, while OSEDiff only needs to fine-tune a small number of LoRA parameters. This makes OSEDiff more memory-efficient and training-friendly. We will cite DMD and discuss its similarity and differences from OSEDiff in the revision.
>
> **Q2. Comparison of the total number of model parameters and FLOPs.**
>
> Thanks for the suggestion. The table below compares the total number of model parameters and FLOPs of different methods.
>
> We see that those pre-trained SD model based methods (i.e., StableSR, DiffBIR, SeeSR, PASD and OSEDiff) have a similar total number of parameters. SinSR employs the diffusion model trained in ResShift, which is much smaller than the pre-trained SD model. However, the generalization capability of ResShift and SinSR is much lower than that of the SD based methods.
>
> In terms of FLOPs, the multi-step diffusion-based models have significantly higher FLOPs than single-step methods (e.g., SinSR and OSEDiff) because they run the SD UNet for multiple times.
> The input resolution of SinSR's VAE decoder is twice of that of OSEDiff's VAE decoder, as SinSR uses an f4 VAE and OSEDiff uses an f8 VAE. Overall, OSEDiff and SinSR have similar FLOPs. We will add these analysis in the revision.
>
> **Table: Complexity comparison among different methods. All methods are tested with an input image of size 512×512 on an A100 GPU.**
>
> |                       | StableSR | DiffBIR | SeeSR | PASD   | ResShift | SinSR | OSEDiff |
> |-----------------------|:--------:|:-------:|:-----:|:------:|:--------:|:-----:|:-------:|
> | **# Total Param (M)** | 1410     | 1717    | 2524  | 1900   | 119      | 119   | 1775    |
> | **FLOPs (G)**         | 79940    | 24234   | 65857 | 29125  | 5491     | 2649  | 2265    |
> | **Inference Step**    | 200      | 50      | 50    | 20     | 15       | 1     | 1       |
> | **Inference Time (s)**| 11.50    | 2.72    | 4.30  | 2.80   | 0.71     | 0.16  | 0.35    |
>
> **Q3. Robust text prompts from LQ.**
>
> Sorry that we did not make it clear. We adopt the DAPE module from SeeSR, which has proven to be robust to the degradation of LQ images for extracting tag-style text prompts. The DAPE module is trained to generate correct semantic prompts even when the input images are severely degraded.
>
> **Q4. Limitations in fidelity.**
>
> Compared with traditional image restoration methods, the recent methods that leverage pre-trained SD priors can produce perceptually much more realistic results. As a compromise, some generated details may not be faithful enough to the input LQ image. This is an inherent limitation of such generative prior based methods, as noted by this reviewer.
>
> Compared with other methods along this line, however, the proposed OSEDiff has achieved a much better balance between fidelity and creativity. First, it takes the LQ image as input without any random noise and performs only one-step diffusion. This significantly improves the stability of image synthesis using diffusion priors, reducing much the possibility of generating false details. On the other hand, OSEDiff utilizes VSD loss to regularize the generator network, ensuring that it can preserve the generative capacity of the pre-trained SD model. As can be seen from the experiments in the main paper, OSEDiff achieves state-of-the-art results in both fidelity and perceptual metrics among the SD based methods. In future research, for image scenes or regions that require high-fidelity, such as face and text, we can apply a higher weight of fidelity loss. For scenes like greenery and hair where the perceptual quality is more important, we can adaptively apply a higher weight of VSD loss to enhance expressiveness.

---

> > ### Comment · Reviewer_KDhM · 2024-08-10
> >
> > Thank the authors for the careful reply. While the implementation draws some insights from ProlificDreamer and DMD, this paper is the first to address super-resolution tasks, and its contribution is valuable to the community. However, I have a further question regarding the VSD loss. Although the ablation experiments demonstrate its effectiveness, I have concerns. In super-resolution, the output must add details and remain consistent with the input. If the noise added in VSD loss is minimal, denoising might yield results similar to the original, making it hard to enhance details. On the other hand, too much noise could increase detail but compromise fidelity. Balancing these factors is challenging, which makes VSD more effective for generation tasks but less theoretically clear for image restoration. Could the authors offer a more illustration about this point? Would using a larger super-resolution model for distillation be a more reasonable approach?

---

> > > ### Author Response · Authors · 2024-08-11
> > >
> > > **Q1. How to balance perception and fidelity in OSEDiff?**
> > >
> > > We sincerely thank this reviewer for the recognition on our contribution. This reviewer's comments on the role of VSD loss in OSEDiff is correct. VSD was originally proposed for the 3D generation task. We adopt it as a regularization term on image distribution, aiming to generate details and enhance the naturalness of the SR output. Adding heavier noise in VSD loss will improve generative ability, while adding lighter noise will result in better fidelity. In our implementation, we uniformly sampled $t$ in the VSD loss from 1 to 1000 in each iteration. Kindly note that in addition to VSD loss, we also employed the $L_2$ loss and LPIPS loss as fidelity constraints in OSEDiff. We balance perceptual quality and fidelity by adjusting the weights on different loss terms. We empirically set the weights of $L_2$ loss, LPIPS loss, and VSD Loss as 1, 2 and 1, respectively, and found that a good balance on perceptual quality and fidelity can be achieved, as evidenced by the experiments in our manuscript.
> > >
> > > **Q2. Would using a larger super-resolution model for distillation be a more reasonable approach?**
> > >
> > > Thanks for the insightful comments. Actually, at the very beginning of this work, we indeed have tried to use a large diffusion-based SR model, more specifically SeeSR, as the regularization model to distil OSEDiff. However, the trained model suffered from weak detail generation ability. This is because SeeSR, as an image restoration model but not a generation model, is already a trade-off between fidelity and perception. Its generative ability is reduced to ensure the fidelity of SR outputs. When using SeeSR as the regularization model, it is difficult to endow the OSEDiff model enough capacity to generate image details. Therefore, we directly use a pre-trained SD model as the regularization model.

---

> > > > ### Comment · Reviewer_KDhM · 2024-08-13
> > > >
> > > > I acknowledge that balancing fidelity and perceptual quality is challenging and hope the authors will provide more experiments and discussions on this issue. Thank the authors for addressing my concerns. I would like to raise my score.

---

### Official Review · Reviewer_jXQz · 2024-07-14

**Soundness:** 3
**Presentation:** 3
**Contribution:** 3
**Rating:** 6
**Confidence:** 4

**Summary:**

The paper introduces OSEDiff, a one-step diffusion network for Real-World Image Super-Resolution. OSEDiff performs variational score distillation in the latent space to ensure predicted scores align with those of multi-step pre-trained models. ​ By fine-tuning the pre-trained diffusion network with trainable LoRA layers, OSEDiff can adapt to real-world image degradations. Experimental results show that OSEDiff achieves comparable or superior Real-ISR outcomes to previous methods. ​ The paper's contributions include the development of OSEDiff, which addresses the limitations of traditional multi-step diffusion models and enables efficient production of high-quality images in one diffusion step. ​

**Strengths:**

1.	The proposed method outperforms the previous single-step SR method (SinSR) in terms of the most of metrics.
2.	It is comparable to previous multi-step diffusion models but greatly reduces inference time.
3.	The model has very few training parameters and can be trained with minimal resources, which is advantageous.
4.	Figure 3 shows that the qualitative comparison results are very promising.

**Weaknesses:**

1.	How much would the performance of OSEDiff change if multiple denoising steps were performed, such as 20 steps or 50 steps?
2.	The design of the key module, the regularizer networks, is not clearly explained.
3.	It is unclear how the LR image is used as a condition. Is it concatenated along the channel dimension in the latent space? However, the pre-trained SD’s VAE encoder downscales by a factor of eight, so for a 512x512 HR input, the latent output is only 64x64, which conflicts with the 128*128 LQ setting mentioned in the first paragraph of section 4.1.
4.	The reason for fine-tuning the VAE encoder is not very clear. Please explain in the revision why it is necessary to fine-tune the VAE encoder jointly with the diffusion model. If this has already been discussed, please indicate where it is covered.
5.	Does the model support input of LR images with different resolutions? For example, if the input is a 32x32 LR image, can feasible results be obtained?

**Questions:**

Please refer to the weakness part

---

> ### Author Rebuttal · Authors · 2024-08-06
>
> **Q1. Multiple inference steps for OSEDiff.**
>
> Please kindly note that OSEDiff is specifically designed for one-step diffusion for Real-ISR. Unlike  previous multi-step methods (e.g., StableSR, PASD, SeeSR), which are all based on ControlNet by using noise as input and LQ image as control signal, OSEDiff directly uses LQ image as input (without random noise) and apply LoRA to finetune the diffusion network for producing HQ output. In its current network design, OSEDiff cannot be used for multiple denoising steps. Nonetheless, we believe that for the Real-ISR task, OSEDiff is much preferred as it achieves faithful and perceptually realistic results while significantly reducing the complexity. If we wish to extend OSEDiff to a multi-step framework for stronger generative capability, we could model multi-step diffusion in the residual space, similar to ResShift. We will explore this possibility in the future.
>
> **Q2. Details about regularizer networks.**
>
> Thanks for pointing out this problem. We will add more details on the regularizer networks in the revision. To model the distribution of natural and restored images, two diffusion models are needed to compute the KL-divergence via variational score distillation (VSD): one for the natural image distribution and another for the restored image distribution. Since the pre-trained SD model can effectively represent the natural image distribution, one regularizer network can be the frozen SD U-net. On the other hand, the U-net can be fine-tuned to model the distribution of restored images. Therefore, we fine-tune the SD U-net as another regularizer network in a similar way to that we fine-tune the SD as the generator network. However, they are trained differently: the generator is trained on $t=999$ to perform super-resolution, while the regularizer is trained on $t \in (\{0, \dots,999\})$ to align with the restored image distribution via VSD loss.
>
> **Q3. LQ as condition.**
>
> Sorry for the confusion caused. Actually, different from previous methods such as StableSR, PASD and SeeSR, our proposed OSEDiff does not use the LQ image as a condition but directly as the input to the UNet. First, we upsample the LQ image to the target size. Then, we input it into the encoder to obtain the latent feature, which is passed into the UNet to obtain the refined feature. Finally, the refined feature goes through the decoder to output the restored HQ image. For example, for $\times$4 SR on a 128$\times$128 LQ image, we first use a bicubic interpolator to upsample it to 512$\times$512. Then, we pass it to OSEDiff to get a 512$\times$512 HQ image. We will clarify this in the revised manuscript.
>
> **Q4. Finetune VAE encoder.**
>
> We apologize for any lack of clarity in our manuscript. The VAE encoder is set to be trainable to remove degradation. In other words, we fine-tune the encoder with LoRA so that it can also serve as a degradation removal network to pre-process the input LQ image.
>
> To better illustrate the role of fine-tuning the encoder, we compare the performance of OSEDiff on the RealSR dataset with and without fine-tuning the VAE encoder. The results are shown in the table below. We can see that fine-tuning the encoder significantly improves the no-reference metrics. Although fine-tuning the encoder leads to slightly worse full-reference metrics such as PSNR and LPIPS, these metrics do not necessarily indicate better visual quality. In Figure 1 of the enclosed PDF file, we visualize the results of OSEDiff with and without encoder fine-tuning. One can see that fixing the VAE encoder may introduce some artifacts, which can be caused by the severe degradation in the LQ input. We will add the above discussions in the revision.
>
> **Table: Comparison of OSEDiff with and without fine-tuning the VAE encoder on the RealSR dataset.**
> |               | Finetune VAE Encoder | PSNR↑ | SSIM↑  | LPIPS↓ | CLIPIQA↑ | MUSIQ↑ | NIQE↓  |
> |---------------|:-----------------:|:-----:|:------:|:------:|:--------:|:------:|:------:|
> | OSEDiff       |        ✗          | 25.27 | 0.1966 | 0.2656 | 0.5303   | 58.99  | 6.5496 |
> | OSEDiff       |        ✓          | 25.15 | 0.2128 | 0.2921 | 0.6693   | 69.09  | 5.6479 |
>
> **Q5. LR inputs of different resolutions.**
>
> Yes, OSEDiff can handle inputs of different resolutions in a similar way to other diffusion based super-resolution methods. First, the LR image (with the size of H$\times$W) is upsampled to an HR image $Y_{bicubic}$ (with the size of rH$\times$rW) using bicubic interpolation, where $r$ is the SR factor. If the short side of $Y_{bicubic}$ is less than 512, we resize its short side to 512 before feeding it to OSEDiff. Finally, the output is resized back to rH$\times$rW to obtain $Y_{osediff}$. In other cases, $Y_{bicubic}$ (rH$\times$rW) is directly fed into OSEDiff to get $Y_{osediff}$ (rH$\times$rW). For the $\times$4 SR task, if LR is 32$\times$32, we first upsample LR to 512$\times$512 using bicubic interpolation, enhance it to  512$\times$512 by OSEDiff, and then resize the output to 256$\times$256.

---

> > ### Comment · Reviewer_jXQz · 2024-08-09
> >
> > Thanks for the response. The authors have solved my problems and I am about to maintain my positive recommendation; hope to see the VSD training code soon.

---

> > > ### Author Response · Authors · 2024-08-09
> > >
> > > We sincerely thank this reviewer for the positive feedback. For sure we will release the training codes and all the trained models soon.
> > >
> > > Authors of paper 4246

---

### Author Rebuttal · Authors · 2024-08-06

Dear Reviewers, Area Chairs, and Program Chairs:

We are grateful for the constructive comments and valuable feedback from the reviewers. We appreciate the reviewers' recognition on the novelty of our method (Reviewers KDhM and 4vqL), its superior performance (Reviewers jXQz and px3F), efficient training (Reviewer jXQz), and its clarity (Reviewers KDhM, px3F, and 4vqL).

To address the reviewers' concerns, we have provided more analysis about inference setting, VAE encoder finetuning, the distribution of output, comparison with DMD and SinSR, and the total parameters and FLOPs. We have also attached a PDF file for more visual comparisons. Please find our itemized responses to all reviewer’s comments below. We will really appreciate it if Reviewer px3F can kindly reconsider the decision, provided that the main concerns are well addressed.

Best regards,

Authors of Paper 4246

---

### Decision · Program_Chairs · 2024-09-25

**Decision:**

Accept (poster)

**Comment:**

This manuscript proposes a single-step image super-resolution model based on diffusion models, utilizing DMD loss for distillation, which results in faster generation compared to multi-step methods. All reviewers have provided positive or neutral scores, so the final decision for this manuscript is acceptance.

However, the manuscript has issues with insufficient citation of prior work. The proposed method, which employs LoRA finetuning, heavily relies on 'One-Step Image Translation with Text-to-Image Models' [1], which demonstrates the use of Stable Diffusion as a base model with LoRA finetuning, yet this work is not cited in the manuscript. The proposed loss function is essentially the DMD loss, but DMD [2] is not cited either. Additionally, contrary to what was mentioned in the authors' rebuttal, the use of LoRA for efficient finetuning is already provided in the official DMD code. Moreover, the manuscript bears significant similarity to SwiftBrush [3], but this work is not cited. It is strongly recommended that the authors cite the aforementioned prior works and include a discussion of them in the revised version of the manuscript.

Furthermore, there are concerns regarding the fairness of the experimental setup. While the proposed method and SeeSR[4] were trained on the high-quality LSDIR [5] dataset, other comparison methods used different datasets. This discrepancy in training data, particularly given the superior quality of LSDIR, leads to an unfair advantage and potentially unreliable comparison results. It is strongly recommended that the authors additionally provide experimental results using the same training datasets as the other methods.

[1] Parmar, Gaurav, et al. "One-step image translation with text-to-image models." arXiv preprint arXiv:2403.12036.

[2] Yin, Tianwei, et al. "One-step diffusion with distribution matching distillation." CVPR. 2024.

[3] Nguyen, Thuan Hoang, and Anh Tran. "Swiftbrush: One-step text-to-image diffusion model with variational score distillation." CVPR 2024.

[4] Wu, Rongyuan, et al. "Seesr: Towards semantics-aware real-world image super-resolution."CVPR 2024.

[5] Li, Yawei, et al. "Lsdir: A large scale dataset for image restoration." CVPR 2023.